# Economic freedom index effects on inbound tourism in European countries: A spatial analysis

Sakar Hasan Hamza[1]*, Qingna Li[1], Mohsen Khezri[2]

**1** School of Mathematics and Statistics, Beijing Institute of Technology, Beijing, China, **2** Department of Geography and Environment, London School of and Political Science (LSE), London, United Kingdom

* sakar.hasan16@yahoo.com

## Abstract

Despite the significance of economic freedom in tourism dynamics, especially from a spatial standpoint, its nuanced influence remains unexplored mainly in current research. To fill this gap, our study introduces a novel spatial panel data analysis to investigate how various components of the economic freedom index affect tourist arrivals in 41 European countries from 2005 to 2018. By employing this innovative approach, we uncover the complex interdependencies between economic freedom and tourism and highlight the significance of regional economic characteristics on the tourism sector's health. Our findings reveal that a one percent increase in GDP per capita of neighboring nations corresponds to a 0.4 percent increase in tourist arrivals to the home country. In comparison, a similar rise in neighboring countries' prices leads to a 0.4 percent decrease in inbound tourists. Most economic freedom variables, including the Business Freedom Index, Investment Freedom Index, Labor Freedom Index, Trade Freedom Index, and Government Integrity Index, demonstrate statistically significant positive effects. However, a one percent increase in the Monetary Freedom Index of neighboring countries results in a 0.747 percent reduction in homebound tourists. Notably, enhancements in the country's and neighboring countries' Investment Freedom Index and Government Integrity Index contribute to increased arrivals. This research contributes to the broader understanding of economic policies' impact on tourism, offering valuable insights for policymakers aiming to leverage economic freedom for tourism development. The application of a spatial panel data approach marks a significant methodological advancement in tourism studies, opening new avenues for analyzing economic influences on tourism at a regional level.

## Introduction

Europe is a crucial player within the global economic tableau, especially in the tourism sector. This sector generated an estimated $2.191 trillion in direct and indirect revenues in 2019. This substantial figure equated to approximately 10.3% of the continent's Gross Domestic Product (GDP) and bolstered 11.7% of its total employment in 2018 [1]. Such statistics underscore the critical importance of tourism in fostering economic development and job creation. Within

**Data availability statement:** All relevant data are within the paper and its supporting information files.

**Funding:** The author(s) received no specific funding for this work.

**Competing interests:** The authors have declared that no competing interests exist.

this framework, the concept of economic freedom emerges as a fundamental driver that shapes the expansion and resilience of the tourism industry. Economic freedom encompasses the fundamental right of individuals and businesses to control their labor and property in an environment where a fair legal system protects their rights. This concept serves as the cornerstone for understanding the intricate relationship between regulatory policies, market dynamics, and the overall vitality of the tourism sector. By setting this definitional groundwork early in our exploration, we establish a clear conceptual framework that guides our investigation into how economic freedom influences tourism development and economic prosperity across European nations. The symbiotic relationship between economic freedom and the advancement of tourism is paramount for dissecting the mechanisms that promote economic flourishing and enhance societal welfare. This is particularly relevant in Europe, where the average score on the Economic Freedom Index is 68.0, markedly surpassing the global mean [2].

Over an extended span, the paramount macroeconomic goal of nations and governing bodies heavily reliant on the tourism sector has been centered around economic expansion, strategically addressing persistent issues such as inadequate employment opportunities [3]. Simultaneously, a multitude of empirical investigations have firmly established the pivotal role played by the tourism industry in fostering economic well-being [4–7]. These assertions find validation in theories like the tourism-led growth theory. Consequently, scholars and development theorists find themselves compelled to gain a profound grasp of the primary stimulants behind the expansion of tourism. Their efforts have encompassed a comprehensive exploration of various economic, political, institutional, and infrastructural factors on both the demand and supply sides. Some of considered variables are as follow: GDP, price level, currency exchange rate, travel cost, travel risk, tourism anxiety, political instability, heritage sites, cultural similarities, and corruption.

Acemoglu et al. [8] argue that factors such as economic liberty and strong institutions are crucial for increasing tourism demand and guiding tourism development toward economic prosperity. Additionally, research indicates that both formal and informal institutions shape economic behavior and influence growth, inequality, and resource allocation [9]. Strong institutions, like property rights, are essential for economic development, while weak institutions can hinder growth. Enhanced economic freedom and institutional quality have been shown to significantly improve the efficiency of the tourism sector, with countries adopting open markets and deregulation strategies often experiencing substantial growth in their tourism industries. As per the Heritage Foundation's elucidation, economic freedom embodies the essence of a liberal economic system, signifying that individuals possess the autonomy to engage in a wide array of economic pursuits safeguarded by legal rights. Anticipations arise that economies characterized by freedom will nurture a more competitive environment, compelling entrepreneurs to deliver superior-quality tourism goods and services [2]. The Heritage Foundation defines four critical categories of economic freedom: rule of law, emphasizing fair and consistent legal systems; market openness, encouraging global trade and investment; regulatory efficiency, aiming for streamlined business operations; and government size, advocating limited intervention to empower individual and market decisions. Furthermore, it serves as a definitive indicator for investors and observers, marking the opportune moment for commencing new business ventures. The detrimental consequences of low economic freedom metrics on the prospects of tourism destinations and their erosion of brand values are underscored by Ağazade [10].

As numerous nations progress toward further integration into the global economy and embrace market-oriented policies, the influence of the economic freedom index on tourism has emerged as a pivotal area of investigation. While existing research has extensively examined factors such as press and personal freedom, civil liberties [11], economic and political freedom [2], and the interplay between institutional quality and corruption [12,13],

a thorough analysis focusing on the individual components of the economic freedom index and their impact on the tourism industry is notably lacking. Consequently, this study fills this gap by examining how these index components affect tourist arrivals in European countries, which are universally recognized as leading tourist destinations. This exploration aims to provide a nuanced understanding of the dynamics at play between economic freedom and tourism, highlighting the significance of these factors in shaping the attractiveness and competitiveness of European destinations in the global tourism market.

In addition, a prevailing perspective among researchers posits the global system as interdependent, where economic advancements in one nation can reverberate significantly across neighboring nations [14–16]. Within this framework, spatial characteristics of factors influencing tourism growth have garnered emphasis, accentuating the potential for regions to benefit from positive spatial influences stemming from thriving tourism sectors in adjacent areas [17–21]. For instance, Romão & Saito [21] underscore the inherently location-centric nature of tourism, featuring frequent interregional flows. As such, global and local indicators of spatial autocorrelation emerge as valuable tools for discerning distinct patterns in regional tourism dynamics. Despite the oversight in previous research regarding the significance of regional influences on variables impacting the tourism sector, the primary objective of this study is to delve into how economic indicators influence tourist arrivals through the lens of a spatial model. This approach enables an examination of how independent variables within a specific country can have repercussions on the tourism industry across different nations. By adopting this perspective, the study aims to shed light on the interconnectedness of economic factors and their collective impact on global tourism trends. Despite the acknowledged importance of the tourism sector and its connection to economic freedom, there remains a conspicuous gap in existing research regarding the direct impact of the various components of the Economic Freedom Index on tourism. Previous studies have extensively explored related dimensions such as civil liberties and political freedom but have often overlooked the comprehensive effects of economic freedom components on tourism growth and sustainability. This gap hinders a holistic understanding of the mechanisms through which economic freedom fosters tourism development, an issue this study aims to address by examining the specific influence of these components on tourist arrivals in European nations.

In essence, this paper makes significant contributions in two principal domains: firstly, it offers a comprehensive examination of the impact of economic freedom on the tourism industry, covering a wide range of economic freedom index components. Secondly, while numerous studies have employed spatial econometrics modeling to analyze the determinants of tourism [22–30], this study is groundbreaking in its application of this methodology to investigate the effects of individual components of the economic freedom index on tourism. This methodology is spurred by the observation that conventional panel econometric frameworks often overlook spatial dynamics. A spatial econometric model is designed to analyze data with spatial dependencies, accounting for spillover effects where an outcome in one area impacts nearby regions [31]. It includes models like the spatial lag model (SLM) and spatial error model (SEM) to capture these spatial relationships, offering insights into inter-regional interactions. Neglecting spatial spillovers in models of regional tourism growth, as noted by Fingleton & López-Bazo [32] and LeSage & Pace [31], can lead to biased and potentially misleading outcomes. Moreover, these approaches fail to consider the nuanced ramifications of the economic freedom index, including its indirect repercussions stemming from neighboring regions, along with the intricate spatial spillover influences on the evolution of tourism. Hence, using spatial econometric models proves notably more beneficial, exhibiting enhanced efficiency and effectiveness in this context. The objectives of this research are twofold. First, to meticulously analyze how different facets of economic freedom—ranging from regulatory

efficiency to market openness—affect the tourism industry in Europe, thereby contributing to a richer understanding of its dynamics and potential for growth. Second, to pioneer spatial econometric models in this realm, this study intends to explore the spatial spillover effects of economic freedom on tourism, highlighting how economic policies in one country can influence tourism trends in neighboring countries. Through this innovative approach, the research aims to unveil the intricate interdependencies within the global tourism landscape, providing actionable insights for policymakers and stakeholders to enhance the attractiveness and competitiveness of European destinations.

This paper follows a segmented structure. Commencing with an introductory section in Part 1, Part 2 delves into an extensive literature review. Part 3 outlines the spatial econometrics model, while Parts 4 and 5 present the estimation results and conclusions.

## Literature review

The evolution of tourism research has underscored the multifaceted significance of diverse factors encompassing economic, political, institutional, infrastructural, and regional dimensions in driving tourism growth. Within this intricate web of determinants, economic factors within the destination country and its neighboring regions have consistently emerged as pivotal contributors to the expansion of tourism. This literature review delves into the chronology and themes within this realm, highlighting seminal studies and their findings.

### Economic factors

The exploration of tourism growth through the lens of academic research has revealed the critical importance of various factors, ranging from economic to regional influences. This literature review aims to provide a more structured and focused analysis, pinpointing critical studies' central themes and contributions within the economic dimension, which is identified as a primary catalyst for tourism expansion. Economic determinants have been consistently highlighted as crucial to the growth of tourism. This section synthesizes the core findings of seminal works, emphasizing the pivotal role of specific economic indicators and their impact on tourism dynamics.

**The influence of GDP and economic variables.** A foundational cluster of studies, including those by Cheung and Saha [33], Larsen et al. [34], Lim [35], Saha and Yap [36], and Yap and Saha [37], has established the significant influence of a destination's economic health, as measured by GDP, on its tourism sector's growth. Saha and Yap [36], alongside Rosselló et al. [38], further refine this understanding by examining how trade openness and economic freedom contribute to an attractive tourism environment, suggesting that these economic indicators can serve as barometers for tourism potential.

**Price levels and exchange rates: A dual perspective.** The work of Forsyth and Dwyer [39] introduces an intricate analysis of how relative prices and exchange rates shape tourist decision-making. The currency exchange rate assumes significance here; a more substantial exchange rate favoring the origin country can sway tourism imports, underscoring the delicate balance between relative prices and tourist arrivals [40]. This dual perspective highlights the nuanced relationship between cost considerations and destination choice, with additional insights from Seetanah et al. [41] on how competitive pricing among destinations influences tourist inflows, illustrating the complexity of tourism economics.

**The role of income and price sensitivity.** Income levels emerge as a significant factor in tourism demand, with studies by Lee et al. [42] dissecting the income and substitution effects on tourist preferences. This analysis is enriched by Morrison's [43] counterintuitive findings on the positive correlation between high price levels and tourism demand, underscoring the diverse factors in destination choice. Rugg [44], Morley [45], and Masiero & Nicolau [46]

further contribute to this discourse by highlighting how tourists' income levels mediate the impact of price on tourism choices, thereby mapping the intricate interplay between economic prosperity and tourism patterns. A consensus emerges among numerous researchers that tourists' income levels wield substantial influence on their decision-making process. Income, construed as a financial boundary, assumes heightened importance, especially for middle- and high-income segments, portraying tourism as a normal good [47–52].

In summation, the chronological and thematic exploration of the literature reveals the evolving significance of economic factors in driving tourism growth. As researchers delve more deeply, their focus shifts from general economic conditions to nuanced variables like relative prices, income levels, and their intricate interplay in tourists' decision-making processes. This journey underscores the multifaceted dynamics shaping the symbiotic relationship between economic factors and the ever-evolving realm of tourism.

## Economic freedom

In the sphere of tourism economics, scholars have extensively examined the influence of economic freedom on tourism demand and the overall competitiveness of the tourism industry. Gholipour et al. [53] and Das & DiRienzo [54] underscore economic freedom as a critical factor shaping tourism demand and advocate for its enhancement to bolster industry competitiveness and inbound tourism. However, while previous studies have explored various aspects of economic freedom, a coherent focus on the key points of each section is lacking.

**Economic freedom and tourism demand.** A closer examination of the components of the freedom index reveals nuanced effects on tourism arrivals. For instance, the size of government correlates with improved tourist infrastructures, while tax burden and business freedom foster entrepreneurial activities, enhancing the quality and competitiveness of tourist services. This notion is supported by Easton & Walker [55], Knack & Keefer [56], and Barro & Sala-i-Martin [57], who argue that economically free nations establish efficient labor and product markets, stable monetary and legal systems and open trade, collectively nurturing a more competitive business environment. Moreover, trade freedom emerges as pivotal in diversifying goods and services, thus heightening visitor satisfaction. Conversely, nations with limited economic freedom may expose tourists to volatile social systems, inefficient judiciaries, and corrupt economic landscapes, potentially leading to inconveniences and hazards for travelers [2]. Fostering competitive conditions through a destination's economic freedom encourages tourism service providers to offer superior goods and services at competitive prices, ultimately intriguing tourists.

**Legal system and tourism industry development.** Many scholars have explored the relationship between economic freedom components and tourism industry development within the context of the legal system's influence. Central to this investigation are facets such as property rights, civil rights, contract enforcement, and infrastructure investments. Robust legal systems and institutions safeguard civil and property rights, while effective contract enforcement catalyzes efficiency across economic sectors, positively impacting international tourism [8]. Ozcan et al. [58] emphasize the importance of infrastructure investments for tourism development, which necessitate stability, prompting consideration of the legal system's efficacy and the level of corruption by potential foreign investors [2].

**Spatial interdependencies and economic freedom.** While some studies have explored the impact of economic freedom indices on tourism development, overlooking spatial interdependencies between neighboring countries, others have delved into this nexus. Empirical analyses by Knack & Keefer [56] and Acemoglu et al. [8] investigate whether economic institutions influence the tourism industry's growth potential, revealing a positive

relationship between economic freedom and tourism arrivals. Additionally, Saha & Yap [36] demonstrate the adverse effects of political instability and terrorism on the tourism industry. Balli et al. [59] highlight the significance of institutional quality in tourism destination selection within OECD countries.

**Inbound tourism and economic freedom.**  Ozcan et al. [58] explore the nexus between incoming foreign tourists and economic freedom, reinforcing the correlation between economic liberty, economic expansion, and the global tourism sector. Similarly, Samitas et al. [60] highlight the role of civil liberties in influencing inbound tourism, while Gozgor et al. [61] stress the importance of an efficient legal framework and robust property rights protection in propelling the tourism domain. Moreover, Kubickova [13] evaluates the repercussions of different freedom components on tourism within Central American nations, uncovering the significant influence of labor, monetary, and business freedoms. However, the effects of freedom from corruption and property rights on destination competitiveness exhibit constraints. Aslan et al. [15] establish a bidirectional causality linking tourist influx and economic freedom indicators, particularly noting a detrimental impact within Mediterranean countries. Similarly, Bulut et al. [11] illustrate how the degree of freedom elucidates levels of international tourist arrivals in heavily frequented nations.

**Tourism, economic vulnerability, and growth.**  Recent inquiries delve into the intricate association between tourism, economic vulnerability, and growth. Wang et al. [62] highlight the potential of global tourism in mitigating economic vulnerability, identifying a critical threshold beyond which benefits materialize concerning GDP per capita. Haini et al. [63] explore the moderating role of social globalization in connecting international tourism and economic growth, finding that social globalization directly and indirectly contributes to development through its augmentation of tourism. Furthermore, Kim et al. [64] evaluate the ramifications of a regional visa waiver initiative on tourism and economic growth, emphasizing governments' cooperative endeavors in stimulating tourism development.

• Destination Competitiveness Theory (DCT)

The concept of Destination Competitiveness Theory (DCT) holds a pivotal position within the domain of the tourism industry, furnishing a conceptual structure for grasping and amplifying a destination's allure and competitive stance. In an era where worldwide travel has attained escalating accessibility, destinations engage in a competitive tussle for the attention of voyagers. This underscores the imperative need for a comprehensive understanding of the constituents contributing to their competitive prowess. DCT plunges into the profundities of these constituents, enveloping a broad spectrum of economic, societal, cultural, and ecological dimensions that collaboratively mold a destination's charm. This theoretical construct explicates the strategic methods through which destinations can strategically position themselves for favorability.

At the very essence of DCT lies the recognition that competitiveness transcends mere price evaluations; instead, it entails a multifaceted assessment of a destination's overall desirability. Crouch and Ritchie [65] underscore the diversity of competitiveness, accentuating the significance of elements like natural endowments, infrastructural capabilities, cultural legacy, and service quality in determining a destination's appeal. The theory underscores that the competitiveness of a destination is sculpted through a dynamic interplay between its supply-side factors (comprising infrastructure, services, and resources) and demand-side factors (encompassing tourists' inclinations, motivations, and perceptions) [66]. Numerous dimensions merge to shape a destination's competitiveness, interweaving intricately. The bedrock of tourism, including infrastructure and transportation networks, stands as vital pillars of a destination's competitive stature [67]. An efficient and well-connected infrastructure not only heightens accessibility but also profoundly influences the holistic perception of the

destination. Moreover, intrinsic culture and authenticity occupy a pivotal role [68], with visitors actively seeking bona fide experiences that mirror the unique identity of the destination.

Perpetually innovative and adapting to shifting trends are requisite for destinations to persist in their competitive standing. Buhalis and Costa [69] underscore the essence of harnessing technological strides to augment competitiveness. In the digital epoch, destinations that astutely harness online platforms and social media are poised to secure a competitive vantage by engaging with a broader audience and prospective travelers. Furthermore, DCT acknowledges the role of governmental policies and synergy among stakeholders in buttressing competitiveness. Statutes and regulations that foster a tourism-friendly milieu can significantly influence a destination's allure [70].

To address the identified gaps in the existing literature, this study examines the nuanced relationship between economic freedom and inbound tourism, incorporating recent trends and broader economic implications. While previous research has established economic freedom as a pivotal factor influencing tourism demand by enhancing sector competitiveness through elements like government size, tax policies, entrepreneurial freedom, and trade openness, there remains a lack of comprehensive analysis on how these factors interact with contemporary trends in international tourism (e.g., digital transformation, sustainable tourism practices). Additionally, the role of spatial dependencies among neighboring nations in shaping tourism dynamics has been underexplored.

Recent studies have begun to highlight the importance of institutional quality, political stability, and social globalization in mediating the effects of economic freedom on tourism growth. However, there is insufficient focus on how policy interventions can leverage economic freedom to foster a conducive environment for inbound tourism. This research bridges these gaps by investigating the broader implications of economic freedom on policy-making, demonstrating how robust legal frameworks and effective governance not only attract foreign investment but also enhance the overall tourism infrastructure. By linking economic freedom directly to policy outcomes, this study provides valuable insights for policymakers aiming to optimize economic policies to drive sustainable tourism and economic development.

## Methodology and data

### Methodology

A comprehensive review of empirical and theoretical literature highlights the necessity to enhance the tourism arrival model by incorporating key economic indicators: the logarithm of GDP per capita ($lnGDP$), the logarithm of destination country prices ($lnPRICE$), and the logarithm of trade openness ($lnOPE$) [18,36,38]. These variables are transformed using natural logarithms to linearize relationships and mitigate skewness, ensuring more reliable coefficient estimates. Specifically, $lnGDP$ captures the economic prosperity and potential spending power of tourists, $lnPRICE$ reflects the cost of tourism-related goods and services in the destination country, and $lnOPE$ measures the country's integration into the global economy by representing the ratio of total trade to GDP. Additionally, the logarithm of the Environmental Freedom Index ($lnEFI$) is included to assess how environmental policies and regulations influence tourist perceptions and decisions. Data processing involves standardizing tourist arrival figures by converting them to a per capita basis, achieved by normalizing tourist arrivals with the destination country's population, allowing for accurate comparisons across countries with varying population sizes. The model is specified as:

$$lnTOUR_{it} = \beta_1 + \beta_2 lnGDP_{it} + \beta_3 lnPRICE_{it} + \beta_4 lnOPE_{it} + \beta_5 lnEFI_{it} + c_i \left(optional\right) + \alpha_t \left(optional\right) + \upsilon_{it}$$

$$(1)$$

Due to the spatial nature of the effects of neighboring country variables on arrivals, three spatial panel data models can be used to examine such spillovers: 1) a lagged dependent variable, 2) a spatially autoregressive process in the error term [71], and 3) the spatial Durbin model, which incorporates the effects of a neighboring country's independent variable on the dependent variable in a specific country [31]. The spatial lag model is formulated as follows:

$$y_{it} = \lambda \sum_{j=1}^{N} w_{ij} y_{jt} + \varphi + x_{it}\beta + c_i \left(optional\right) + \alpha_t \left(optional\right) + \upsilon_{it} \tag{2}$$

where $x_{it}$ for countries $i = 1, ..., N$ at periods $t = 1, ..., T$ denotes a $1 \times K$ vector of independent variables. Additionally, $\beta$ denotes a vector of parameters. $y_{it}$ represents the dependent variable. $\sum_{j=1}^{N} w_{ij} y_{jt}$ denotes the effect of the dependent variables $y_{jt}$ in neighboring countries on the dependent variable $y_{it}$ in a specific country, and $\lambda$ denotes the corresponding parameter. $w_{ij}$ is the $i, j-th$ element of weights matrix $w$. Before matrix standardization, the $i, j-th$ element will have the value of one of two neighboring countries or zero if they are not neighbors. $c_i$ are country-specific intercepts that capture heterogeneity across countries, while $\alpha_t$ are period-specific intercepts that capture heterogeneity across periods.

The omission of these two latter variables could bias the estimates in a cross-sectional and time-series study, respectively [72]. $\upsilon_{it}$ denotes the random error term.

The spatial error model consists of an error term for the unit $i$, $u_{it}$, which is dependent on the error terms for neighboring countries j, $u_{jt}$, a spatial weights matrix W, and a distinctive component $\upsilon_{it}$:

$$y_{it} = \lambda \sum_{j=1}^{N} w_{ij} y_{jt} + \varphi + x_{it}\beta + c_i \left(optional\right) + \alpha_t \left(optional\right) + u_{it}$$

$$u_{it} = \rho \sum_{j=1}^{N} w_{ij} u_{jt} + \upsilon_{it} \tag{3}$$

The spatial Durbin model also incorporates spatially lagged independent variables into the spatial lag model:

$$y_{it} = \lambda \sum_{j=1}^{N} w_{ij} y_{jt} + \varphi + x_{it}\beta + \sum_{j=1}^{N} w_{ij} x_{jt}\theta + c_i \left(optional\right) + \alpha_t \left(optional\right) + \upsilon_{it} \tag{4}$$

where $\sum_{j=1}^{N} w_{ij} x_{jt}$ examines the interaction effect of independent variables $x_{ijt}$ in neighboring countries on the dependent variable $y_{it}$ in a particular country. Additionally, $\theta$ is a $K \times 1$ vector of parameters.

## Data

Tables 1 and 2 summarize the data's constructed variables and descriptive statistics. The authors used data from 41 European countries including Albania, Armenia, Austria, Azerbaijan, Belarus, Belgium, Bosnia and Herzegovina, Bulgaria, Croatia, Cyprus, Czech Republic, Denmark, Estonia, Finland, France, Georgia, Germany, Greece, Hungary, Iceland, Ireland, Italy, Kazakhstan, Latvia, Luxembourg, Moldova, Netherlands, North Macedonia, Norway, Poland, Portugal, Romania, Russia, Slovenia, Spain, Sweden, Switzerland, Turkey, Ukraine, United Kingdom, Malta, from 2005 to 2018 to model the spatial effects of tourism arrivals determinants. A commonly used test, "Moran's I," was employed to examine such spillover effects thoroughly. The computed statistic elucidates the spatial effects depicted in Fig 1. Two

**Table 1. Variable's construction.**

| Variable | Variable constructed | Source |
|---|---|---|
| $lnTOUR_{it}$ | $lnTOURISM_{it} = log(TOURISM_{it})$ <br> $RENEW_{it}$ = Total Tourist arrival per capita | UNWTO |
| $lnGDP_{it}$ | $lnGDPP_{it} = log(GDPP_{it})$ <br> $GDP_{it}$ = GDP per capita (in 2010 prices$) | WDI |
| $lnPRICE_{it}$ | $lnPRICE_{it} = log(PRICE_{it})$ <br> $PRICE_{it}$ = the ratio of PPP conversion factor to the market exchange rate | WDI |
| $lnOPE_{it}$ | $lnOPE_{it} = log(OPE_{it})$ <br> $OPE_{it}$ = Trade Openness (the sum of exports and imports as a percentage of GDP) | WDI |
| $lnEFI_{it}$ | $lnEFI_{it} = log(EFI_{it})$ <br> $EFI_{it}$ . = Total Economic Freedom Index | Heritage |
| $lnBFI_{it}$ | $lnBFI_{it} = log(BFI_{it})$ <br> $BFI_{it}$ Business Freedom Index | Heritage |
| $lnLFI_{it}$ | $lnLFI_{it} = log(LFI_{it})$ <br> $LFI_{it}$ = Labor Freedom Index | Heritage |
| $lnLMF_{it}$ | $lnLMF_{it} = log(LMF_{it})$ <br> $LMF_{it}$ . = Monetary Freedom Index | Heritage |
| $lnTFI_{it}$ | $lnTFI_{it} = log(TFI_{it})$ <br> $TFI_{it}$ Trade Freedom Index | Heritage |
| $lnIFI_{it}$ | $lnIFI_{it} = log(IFI_{it})$ <br> $IFI_{it}$ = Investment Freedom Index | Heritage |
| $lnFFI_{it}$ | $lnFFI_{it} = log(FFI_{it})$ <br> $FFI_{it}$ = Financial Freedom Index | Heritage |
| $lnPRI_{it}$ | $lnPRI_{it} = log(PRI_{it})$ <br> $PRI_{it}$ = Property Rights Index | Heritage |
| $lnGII_{it}$ | $lnGII_{it} = log(GII_{it})$ <br> $GII_{it}$ = Government Integrity Index | Heritage |
| $lnTBI_{it}$ | $lnTBI_{it} = log(TBI_{it})$ <br> $TBI_{it}$ = Tax Burden Index | Heritage |
| $lnGSI_{it}$ | $lnGSI_{it} = log(GSI_{it})$ <br> $GSI_{it}$ = Government Spending Index | Heritage |

Trade Freedom, Investment Freedom, Property Rights, and Government Integrity.

dimensions of Fig 1 are the regional observations of variables and the associated spatial lag data. Positive Moran's I values in Quadrants I and III confirm the spatial accumulation of similar values throughout the region.

Furthermore, a negative Moran's I can be configured in Quadrants II and IV. According to the fitting lines, positive autocorrelation dominates for most countries, indicating that neighboring countries have comparable tourist arrivals and economic freedom indexes, particularly for the latter. As a result, spatial econometric models must be used to examine the effects of determinants of tourist arrivals.

**Table 2. Descriptive statistics from 2005 to 2018.**

| Variable | Mean | Median | Maximum | Minimum | Std. Dev. | Observations |
|---|---|---|---|---|---|---|
| $lnTOUR_{it}$ | 13.776 | 13.916 | 16.480 | 9.950 | 1.197 | 574 |
| $lnGDP_{it}$ . | 9.817 | 9.993 | 11.626 | 7.631 | 1.030 | 574 |
| $lnPRICE_{it}$ | -0.365 | -0.335 | 0.518 | -1.628 | 0.478 | 574 |
| $lnOPE_{it}$ . | 4.574 | 4.517 | 6.012 | 3.816 | 0.435 | 574 |
| $lnEFI_{it}$ | 4.189 | 4.207 | 4.414 | 3.807 | 0.127 | 574 |
| $lnBFI_{it}$ . | | 4.332 | 4.605 | 3.656 | 0.171 | 574 |
| $lnLFI_{it}$ | 4.096 | 4.101 | 4.605 | 3.434 | 0.234 | 574 |
| $lnLMF_{it}$ | | | 4.519 | 3.523 | 0.105 | 574 |
| $lnTFI_{it}$ | 4.426 | 4.459 | 4.500 | 3.789 | 0.077 | 574 |
| $lnIFI_{it}$ | | | 4.554 | 2.303 | 0.379 | 574 |
| $lnFFI_{it}$ | 4.099 | 4.094 | 4.500 | 2.303 | 0.361 | 574 |
| $lnPRI_{it}$ | 4.025 | 4.243 | 4.554 | 2.303 | 0.494 | 574 |
| $lnGII_{it}$ | 3.933 | 3.970 | 4.575 | 2.890 | 0.449 | 574 |
| $lnTBI_{it}$ | 4.224 | 4.273 | 4.544 | 3.487 | 0.238 | 574 |
| $lnGSI_{it}$ | 3.595 | 3.780 | 4.510 | -0.693 | 0.811 | 574 |

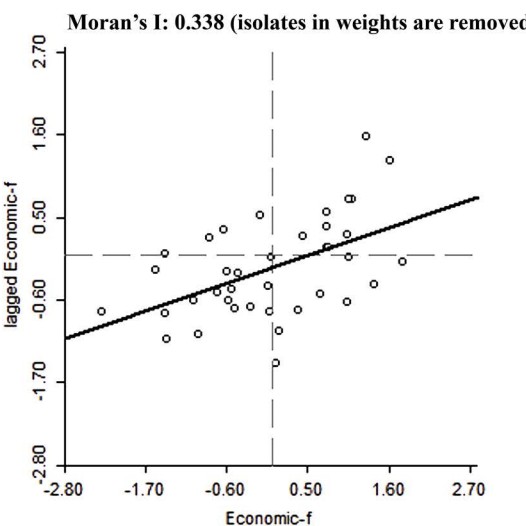
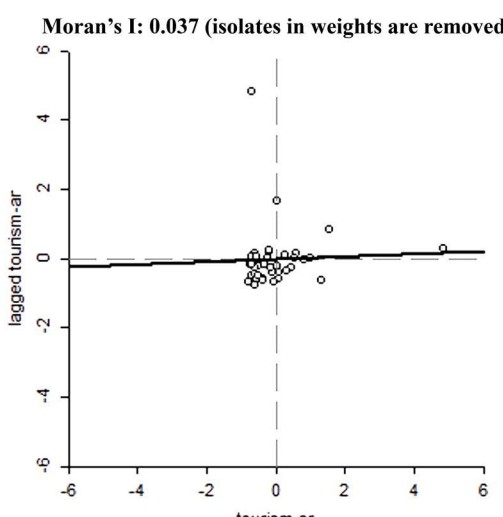

**Fig 1. Moran's I across countries.**

Due to the short duration of some index dimensions, the experimental model has limited to the following indexes: 1) *Business Freedom:* defined as ' 'people's ability to start a business without government interposition. 2) *Labor Freedom:* defined as an individual's ability to find work and a business's ability to contract for labor and release unneeded workers freely. 3) *Monetary Freedom:* defined as the product of the price stability and price control criteria. 4) *Trade Freedom:* defined as the removal of every tariff and non-tariff barrier that imposes

restrictions on international trade. 5) *Investment Freedom:* defined as the absence of restrictions on capital investment stream, allowing firms and individuals to allocate resources between activities freely. 6) *Financial Freedom:* defined as the degree to which individuals are supported in making life choices without being unduly constrained by financial constraints. 7) *Property Rights:* defined as the private sector's ability to accumulate wealth and property freely. 8) *Government Integrity:* defined as practices that protect government institutions from systemic corruption. 9) *Tax Burden:* defined as every kind of direct and indirect tax. 10) *Government Spending:* defined as the total amount of money spent by the government.

The World Development Indicator (WDI), the United Nations World Tourism Organization (UNWTO), and The Heritage Foundation all contributed data to this study (Heritage). Online access to the data is available at https://www.unwto.org/data, www.heritage.org/index, and www.datacatalog.worldbank.org/dataset/world-development-indicators.

## Results

### Diagnostic tests

The outcomes of the diagnostic assessments are in Tables 3–6, located in the article's appendix. Initially, several diagnostic tests were conducted to determine the optimal model. The existence of fixed effects in the model is examined in this study using different likelihood ratio (LR) tests in Table 3. The rejection of the null hypothesis emphasizes the importance of selecting a fixed-effects model with concurrent spatial and time-period effects; otherwise, the following fixed model (i.e., time-period effects or spatial effects) would be chosen. The test results demonstrate the rejection of the null hypothesis and the significance of the LR test statistics in total models.

The LM test is used in Table 4 to determine the presence of spatial interaction effects when the spatial lag or spatial error is included. The LM test's rejection of the null hypothesis confirms the existence of both the spatial lagged and spatial error models. The null hypothesis is rejected for all models involving time and spatial effects. As a result, the model's inclusion of spatial interaction effects emphasizes the importance of considering such effects. The Hausman test in Table 5 examines the possibility of substituting a random-effects model for the fixed-effects model. The null hypothesis confirms the existence of random effects; the test results indicate that random effects are confirmed in the models at a 1% significance level.

In Equation (4), the hypotheses $H_0 : \theta = 0$ for the former and $H_0 : \theta + \lambda\beta = 0$ for the latter are tested to determine whether the spatial Durbin model can be simplified into the spatial lag model and the spatial error model. Table 6 contains the test results. The Wald test statistical values are significant, indicating that the spatial Durbin model cannot be converted to a spatial error or lag model in all cases. As a result, the estimated results are analyzed using a spatial Durbin model with a spatially lagged independent variable.

### Results of estimation

Tables 7 and 8 present the estimation results for various models. To address potential collinearity between multiple indicators of economic freedom, each variable is analyzed in separate, independent models. All models indicate that common control variables significantly influence tourist arrivals. Specifically, each percentage point increase in GDP per capita corresponds to a 1.5 percentage point rise in tourist arrivals. Additionally, trade openness shows a positive and significant effect, with each percentage point increase in trade openness leading to an approximately 0.54 percent increase in tourist arrivals.

Despite the modest statistical significance, the positive impact of rising domestic prices on tourist arrivals deviates from conventional theoretical expectations. Traditionally, higher

**Table 3. The LR test is used to determine whether models contain fixed effects.**

|  | Spatial effects |  | Time-period effects |  |
| --- | --- | --- | --- | --- |
| Model 1 | 80.411 | (0.000)*** | 1875.501 | (0.000)*** |
| Model 2 | 83.087 | (0.000)*** | 1882.232 | (0.000)*** |
| Model 3 | 78.761 | (0.000)*** | 1869.351 | (0.000)*** |
| Model 4 | 81.846 | (0.000)*** | 1874.402 | (0.000)*** |
| Model 5 | 79.682 | (0.000)*** | 1872.735 | (0.000)*** |
| Model 6 | 71.935 | (0.000)*** | 1872.274 | (0.000)*** |
| Model 7 | 66.086 | (0.000)*** | 1865.733 | (0.000)*** |
| Model 8 | 81.562 | (0.000)*** | 1874.699 | (0.000)*** |
| Model 9 | 86.304 | (0.000)*** | 1899.817 | (0.000)*** |
| Model 10 | 87.351 | (0.000)*** | 1897.743 | (0.000)*** |
| Model 11 | 84.725 | (0.000)*** | 1878.683 | (0.000)*** |
| Model 12 | 79.195 | (0.000)*** | 1873.057 | (0.000)*** |

Note: p-values in parentheses, ***, **, and * show significance at the 1%, 5%, and 10% level respectively.

Source: Authors' estimations.

**Table 4. The LM test is used to determine whether models contain spatial lag or spatial error.**

|  |  | Spatial effects |  | Time-period effects |  | Spatial and time-period effects |  |
| --- | --- | --- | --- | --- | --- | --- | --- |
| Model 1 | LM spatial lag | 20.252 | (0.000)*** | 2.370 | (0.124) | 8.300 | (0.004)*** |
|  | LM spatial error | 10.991 | (0.001)*** | 0.078 | (0.780) | 7.878 | (0.005)*** |
| Model 2 | LM spatial lag | 19.192 | (0.000)*** | 2.758 | (0.097)* | 8.091 | (0.004)*** |
|  | LM spatial error | 8.987 | (0.003)*** | 0.162 | (0.687) | 5.678 | (0.017)*** |
| Model 3 | LM spatial lag | 16.097 | (0.000)*** | 5.959 | (0.015)* | 6.730 | (0.009)*** |
|  | LM spatial error | 8.175 | (0.004)*** | 0.906 | (0.341) | 5.778 | (0.016)** |
| Model 4 | LM spatial lag | 20.368 | (0.000)*** | 1.458 | (0.227) | 7.709 | (0.005)*** |
|  | LM spatial error | 11.398 | (0.001)*** | 0.007 | (0.936) | 6.673 | (0.010)*** |
| Model 5 | LM spatial lag | 19.579 | (0.000)*** | 2.864 | (0.091)* | 8.308 | (0.004)*** |
|  | LM spatial error | 10.321 | (0.001)*** | 0.182 | (0.669) | 7.893 | (0.005)*** |
| Model 6 | LM spatial lag | 15.024 | (0.000)*** | 3.333 | (0.068)* | 7.899 | (0.005)*** |
|  | LM spatial error | 6.856 | (0.009)*** | 0.310 | (0.578) | 6.340 | (0.012)** |
| Model 7 | LM spatial lag | 16.203 | (0.000)*** | 3.244 | (0.072)* | 7.804 | (0.005)*** |
|  | LM spatial error | 5.754 | (0.016)** | 0.288 | (0.592) | 5.656 | (0.017)** |
| Model 8 | LM spatial lag | 20.042 | (0.000)*** | 2.123 | (0.145) | 7.941 | (0.005)*** |
|  | LM spatial error | 10.669 | (0.001)*** | 0.002 | (0.968) | 7.097 | (0.008)*** |
| Model 9 | LM spatial lag | 20.122 | (0.000)*** | 1.272 | (0.259) | 9.517 | (0.002)*** |
|  | LM spatial error | 12.441 | (0.000)*** | 0.071 | (0.790) | 7.249 | (0.007)*** |
| Model 10 | LM spatial lag | 17.634 | (0.000)*** | 2.399 | (0.121) | 6.144 | (0.013)** |
|  | LM spatial error | 8.906 | (0.003)*** | 0.087 | (0.768) | 4.406 | (0.036)** |
| Model 11 | LM spatial lag | 20.515 | (0.000)*** | 2.320 | (0.128) | 7.453 | (0.006)*** |
|  | LM spatial error | 11.000 | (0.001)*** | 0.210 | (0.647) | 7.119 | (0.008)*** |
| Model 12 | LM spatial lag | 18.836 | (0.000)*** | 2.712 | (0.100) | 8.586 | (0.003)*** |
|  | LM spatial error | 10.264 | (0.001)*** | 0.074 | (0.786) | 8.021 | (0.005)*** |

Note: p-values in parentheses,***, **, and * show significance at the 1%, 5%, and 10% level respectively.

Source: Authors' estimations.

**Table 5. Hausman test results.**

|  | Spatial lag model |  | Spatial Durbin model |  |
|---|---|---|---|---|
| Model 1 | 86.787 | (0.000)*** | 41.850 | (0.000)*** |
| Model 2 | 177.240 | (0.000)*** | 0.776 | (1.000) |
| Model 3 | 64.670 | (0.000)*** | 36.094 | (0.000)*** |
| Model 4 | 8.790 | (0.118) | 53.537 | (0.000)*** |
| Model 5 | 201.902 | (0.000)*** | 34.723 | (0.000)*** |
| Model 6 | 196.124 | (0.000)*** | 46.395 | (0.000)*** |
| Model 7 | 62.232 | (0.000)*** | 23.744 | (0.005)*** |
| Model 8 | 41.187 | (0.000)*** | 30.083 | (0.000)*** |
| Model 9 | 81.513 | (0.000)*** | 43.392 | (0.000)*** |
| Model 10 | 74.860 | (0.000)*** | 28.858 | (0.001)*** |
| Model 11 | 104.451 | (0.000)*** | 34.572 | (0.000)*** |
| Model 12 | 78.027 | (0.000)*** | 42.087 | (0.000)*** |

Note: p-values in parentheses, ***, **, and * show significance at the 1%, 5%, and 10% level respectively.

Source: Authors' estimations.

**Table 6. A spatial Durbin model compared to a spatial error and lag model.**

|  | Wald test |  |  |  | LR test |  |  |  |
|---|---|---|---|---|---|---|---|---|
|  | Spatial Durbin model and spatial lag model |  | Spatial Durbin model and spatial error model |  | Spatial Durbin model and spatial lag model |  | Spatial Durbin model and spatial error model |  |
| Model 1 | 7.506 | (0.057)* | 19.254 | (0.000)*** | 8.106 | (0.044)** | 20.872 | (0.000)*** |
| Model 2 | 14.398 | (0.006)*** | 26.847 | (0.000)*** | 15.590 | (0.004)*** | 29.111 | (0.000)*** |
| Model 3 | 13.010 | (0.011)** | 23.596 | (0.000)*** | 14.019 | (0.007)*** | 25.327 | (0.000)*** |
| Model 4 | 10.626 | (0.031)** | 21.694 | (0.000)*** | 11.395 | (0.022)** | 23.542 | (0.000)*** |
| Model 5 | 12.907 | (0.012)** | 24.673 | (0.000)*** | 13.884 | (0.008)*** | 26.624 | (0.000)*** |
| Model 6 | 10.107 | (0.039)** | 20.945 | (0.000)*** | 10.947 | (0.027)** | 22.410 | (0.000)*** |
| Model 7 | 37.988 | (0.000)*** | 49.624 | (0.000)*** | 39.999 | (0.000)*** | 53.145 | (0.000)*** |
| Model 8 | 8.400 | (0.078)* | 20.095 | (0.000)*** | 9.050 | (0.060)* | 21.795 | (0.000)*** |
| Model 9 | 5.581 | (0.233) | 15.832 | (0.003)*** | 5.997 | (0.199) | 17.052 | (0.002)*** |
| Model 10 | 10.755 | (0.029)** | 22.349 | (0.000)*** | 11.630 | (0.020)** | 23.691 | (0.000)*** |
| Model 11 | 7.657 | (0.105) | 19.852 | (0.001)*** | 8.307 | (0.081)* | 21.431 | (0.000)*** |
| Model 12 | 9.187 | (0.057)* | 20.141 | (0.000)*** | 9.876 | (0.043)* | 21.907 | (0.000)*** |

Note: p-values in parentheses, ***, **, and * show significance at the 1%, 5%, and 10% level respectively.

Source: Authors' estimations.

prices in the home country are expected to raise the costs of goods and services for tourists, potentially reducing the country's competitiveness. This conventional view highlights the sensitivity of tourist demand to price changes. However, data from 170 countries in 2017, depicted in Fig 2, reveal a counterintuitive positive correlation between price levels and tourist arrivals, suggesting an atypical relationship between these variables.

This anomaly can be partially explained by considering Europe's open economic framework, where price level differences among countries primarily stem from disparities in the costs of non-tradable goods and services, particularly in the service sector. Higher service sector prices often indicate a nation's elevated GDP per capita, reflecting advanced infrastructure and broader inclusivity in welfare and development outcomes. Consequently, European

**Table 7. Models 1 to 6: estimation results.**

|  | Model 1 | Model 2 | Model 3 | Model 4 | Model 5 | Model 6 |
|---|---|---|---|---|---|---|
| *lnGDPP* | 1.533 | 1.435 | 1.458 | 1.540 | 1.514 | 1.370 |
|  | (0.000)*** | (0.000)*** | (0.000)*** | (0.000)*** | (0.000)*** | (0.000)*** |
| *lnPRICE* | 0.185 | 0.200 | 0.200 | 0.188 | 0.211 | 0.195 |
|  | (0.151) | (0.118) | (0.120) | (0.145) | (0.102) | (0.124) |
| *lnOPE* | 0.531 | 0.552 | 0.577 | 0.541 | 0.539 | 0.506 |
|  | (0.000)*** | (0.000)*** | (0.000)*** | (0.000)*** | (0.000)*** | (0.000)*** |
| *lnEFI* |  | 0.286 |  |  |  |  |
|  |  | (0.279) |  |  |  |  |
| *lnBFI* |  |  | 0.131 |  |  |  |
|  |  |  | (0.270) |  |  |  |
| *lnLFI* |  |  |  | -0.012 |  |  |
|  |  |  |  | (0.905) |  |  |
| *lnMFI* |  |  |  |  | -0.027 |  |
|  |  |  |  |  | (0.855) |  |
| *lnTFI* . |  |  |  |  |  | 0.729 |
|  |  |  |  |  |  | (0.000)*** |
| $W \times lnGDPP$ | -0.020 | -0.216 | -0.164 | -0.051 | 0.004 | -0.124 |
|  | (0.909) | (0.255) | (0.373) | (0.776) | (0.984) | (0.492) |
| $W \times lnPRICE$ | -0.356 | -0.331 | -0.392 | -0.380 | -0.399 | -0.388 |
|  | (0.011)** | (0.018)** | (0.005)*** | (0.007)*** | (0.004)*** | (0.005)*** |
| $W \times lnOPE$ | -025 | 0.028 | -0.007 | -0.032 | -0.007 | -0.061 |
|  | (0.888) | (0.871) | (0.967) | (0.854) | (0.967) | (0.724) |
| $W \times lnEFI$ |  | 0.957 |  |  |  |  |
|  |  | (0.012)** |  |  |  |  |
| $W \times lnBFI$ |  |  | 0.341 |  |  |  |
|  |  |  | (0.044)* |  |  |  |
| $W \times lnLFI$ |  |  |  | 0.334 |  |  |
|  |  |  |  | (0.073)* |  |  |
| $W \times lnMFI$ |  |  |  |  | -0.601 |  |
|  |  |  |  |  | (0.021)** |  |
| $W \times lnTFI$ |  |  |  |  |  | 0.231 |
|  |  |  |  |  |  | (0.287) |
| $W \times lnTOUR$ | 0.228 | 0.220 | 0.219 | 0.225 | 0.236 | 0.202 |
|  | (0.000)*** | (0.000)*** | (0.000)*** | (0.000)*** | (0.000)*** | (0.000)*** |

Note: The values in parentheses indicate p-values.

Source: Authors' estimations.

countries tend to show a more pronounced impact of price levels on the supply side of tourism services. This theoretical dissonance is further highlighted by examining the impact of price variability in neighboring countries on home tourist arrivals (captured by the coefficient $W \times lnPRICE$). Each percent increase in neighboring countries' price levels correlates with a

**Table 8. Models 7 to 12: estimation results.**

| | Model 7 | Model 8 | Model 9 | Model 10 | Model 11 | Model 12 |
|---|---|---|---|---|---|---|
| $lnGDPP$ . | 1.355 | 1.549 | 1.459 | 1.403 | 1.541 | 1.523 |
| | (0.000)*** | (0.000)*** | (0.000)*** | (0.000)*** | (0.000)*** | (0.000)*** |
| $lnPRICE$ . | 0.203 | 0.180 | 0.166 | 0.188 | 0.180 | 0.193 |
| | (0.104) | (0.163) | (0.192) | (0.141) | (0.163) | (0.136) |
| $lnOPE$ . | 0.541 | 0.509 | 0.472 | 0.554 | 0.517 | 0.531 |
| | (0.000)*** | (0.000)*** | (0.000)*** | (0.000)*** | (0.000)*** | (0.000)*** |
| $lnIFI$ | 0.7 | | | | | |
| | (0.003)*** | | | | | |
| $lnFFI$ | | .020 | | | | |
| | | (0.789) | | | | |
| $lnPRI$ | | | | | | |
| | | | (0.000)*** | | | |
| $lnGII$ | | | | 0.214 | | |
| | | | | (0.006)*** | | |
| $lnTBI$ | | | | | -0.144 | |
| | | | | | (0.468) | |
| $lnGSI$ | | | | | | -0.004 |
| | | | | | | (0.851) |
| $W \times lnGDPP$ | -0.228 | -0.036 | -0.002 | -0.201 | -0.025 | 0.008 |
| | (0.198) | (0.841) | (0.989) | (0.273) | (0.889) | (0.965) |
| $W \times lnPRICE$ | -0.255 | -0.354 | -0.296 | -0.287 | -0.353 | -0.331 |
| | (0.061)* | (0.011)** | (0.032)** | (0.039)** | (0.012)** | (0.019)** |
| $W \times lnOPE$ | -0.026 | 0.004 | -0.026 | 0.105 | -0.015 | -0.057 |
| | (0.878) | (0.983) | (0.880) | (0.551) | (0.932) | (0.749) |
| $W \times lnIFI$ | 0.648 | | | | | |
| | (0.000)*** | | | | | |
| $W \times lnFFI$ | | 0.116 | | | | |
| | | (0.349) | | | | |
| $W \times lnPRI$ | | | -0.064 | | | |
| | | | (0.486) | | | |
| $W \times lnGII$ | | | | 0.220 | | |
| | | | | (0.072)* | | |
| $W \times lnTBI$ | | | | | 0.100 | |
| | | | | | (0.731) | |
| $W \times lnGSI$ | | | | | | -0.048 |
| | | | | | | (0.176) |
| $W \times lnTOUR$ | 0.159 | 0.227 | 0.230 | 0.221 | 0.233 | 0.220 |
| | (0.001) | (0.000)*** | (0.000)*** | (0.000)*** | (0.000)*** | (0.000)*** |

Note: The values in parentheses indicate p-values.

Source: Authors' estimations.

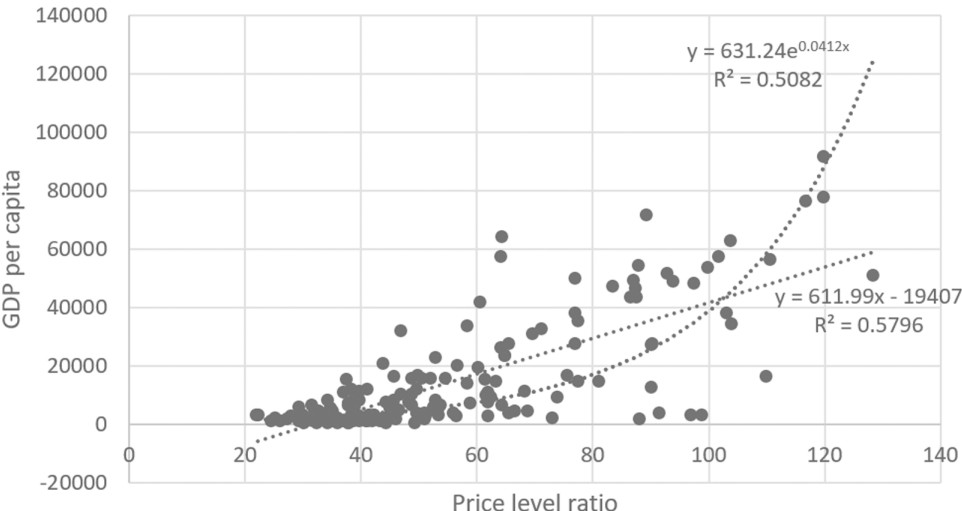

**Fig 2. Positive correlation between tourist arrivals per capita & level of prices.** (Source: WDI).

0.3% reduction in tourist arrivals to the home country, suggesting that price hikes in adjacent nations can divert tourist flows, reducing arrivals to the home nation. This phenomenon underscores the complex interplay between price levels and tourist mobility, challenging traditional views and emphasizing the need for a nuanced understanding of the factors influencing international tourism dynamics.

In Model 2, the total economic freedom variable coefficient is positive with a low significance level. Examining the effects of its constituents reveals that the source of such meaningless effects is the conflicting influences between its constituents. While the six economic freedom indices' coefficients are meaningless, the Trade Freedom Index, Investment Freedom Index, Property Rights Index, and Government Integrity Index are all significantly positive.

Spatial models enable the separation of variables' direct and spillover effects. Direct effects quantify the impact of independent variables on the country's dependent variable. In contrast, spillover effects quantify the impact of independent variables in neighboring countries on a specific country's dependent variable. The spillover and direct impact of the common variables in Model 1 and the economic freedom indices in Models 2-12 are summarized in Table 9.

The empirical evidence delineates a direct correlation between the economic prosperity of neighboring countries and the tourist influx into the home country. Specifically, for every one percent rise in the GDP per capita of neighboring nations, there's a corresponding increase of approximately 0.4 percent in tourist arrivals to the home country. Such a correlation underscores the interconnected nature of economic growth and tourism dynamics across borders. Economic advancement in adjacent countries likely enhances citizens' disposable income, encouraging outbound tourism. Additionally, the regional tourism industry's growth, spurred by economic prosperity, could make the entire area more attractive to international tourists, benefiting neighboring countries, including the home nation. Moreover, this trend reflects a broader pattern observed in the developed world, where increasing wealth and economic development are closely linked with higher international travel and tourism rates.

Furthermore, as reiterated earlier, for every percentage rise in prices in neighboring countries, there will be a corresponding decrease of 0.4 percent in inbound tourists to the home country. The index of neighboring countries' total economic freedom demonstrates positive and significant effects. Furthermore, most economic freedom variables, such as the Business

**Table 9. Tourism determinants' marginal effects.**

| | Direct | | Indirect | | Total | |
|---|---|---|---|---|---|---|
| | Coefficient | p-value | Coefficient | p-value | Coefficient | p-value |
| *lnGDPP* | 1.559 | (0.000)*** | 0.408 | (0.025) | 1.967 | (0.000)*** |
| *lnPRICE* | 0.162 | (0.194) | -0.390 | (0.007)*** | -0.228 | (0.021)** |
| *lnOPE* | 0.542 | (0.000)*** | 0.100 | (0.613) | 0.642 | (0.003)*** |
| *lnEFI* | 0.380 | (0.156) | 1.253 | (0.007)*** | 1.633 | (0.003)*** |
| *lnBFI* | 0.152 | (0.198) | 0.450 | (0.032)** | 0.601 | (0.008)*** |
| *lnLFI* | 0.012 | (0.911) | 0.401 | (0.092)* | 0.412 | (0.158) |
| *lnMFI* | -0.066 | (0.674) | -0.747 | (0.022)** | -0.813 | (0.038)** |
| *lnTFI* | 0.756 | (0.000)*** | 0.448 | (0.076)* | 1.205 | (0.000)*** |
| *lnIFI* | 0.233 | (0.001)*** | 0.774 | (0.000)*** | 1.007 | (0.000)*** |
| *lnFFI* | -0.011 | (0.878) | 0.130 | (0.400) | 0.118 | (0.503) |
| *lnPRI* | 0.251 | (0.000)*** | -0.008 | (0.936) | 0.243 | (0.017)** |
| *lnGII* | 0.233 | (0.004)*** | 0.325 | (0.035)** | 0.558 | (0.002)*** |
| *lnTBI* | -0.146 | (0.460) | 0.069 | (0.845) | -0.077 | (0.852) |
| *lnGSI* | -0.007 | (0.719) | -0.056 | (0.200) | -0.063 | (0.233) |

Note: The values in parentheses indicate p-values.

Source: Authors' estimations.

Freedom Index, the Investment Freedom Index, the Labor Freedom Index, the Trade Freedom Index, and the Government Integrity Index, exhibit a statistically significant positive effect—however, neighboring countries' Monetary Freedom Index results in a decrease in the number of arrivals. Only improvements in the country's and neighboring countries' Investment Freedom Index and Government Integrity Index will increase arrivals among the various indicators. On the other hand, the Financial Freedom Index, the Tax Burden Index, and the Government Spending Index demonstrate no significant effect.

The findings suggest that the overall economic freedom in neighboring countries positively influences domestic economic activities, which aligns with existing literature on regional economic integration and spillover effects. Studies such as Gwartney et al. [73] highlight how higher economic freedom fosters a conducive environment for business operations, thereby boosting economic growth. The positive effects of the Business Freedom Index, Investment Freedom Index, Labor Freedom Index, Trade Freedom Index, and Government Integrity Index corroborate this notion. These indices reflect a country's regulatory efficiency, investment climate, labor market flexibility, trade openness, and integrity in governance—all crucial for creating a stable and attractive economic environment that can stimulate domestic economic activities, including tourism and investment.

Contrarily, the Monetary Freedom Index showing a decrease in domestic arrivals may seem paradoxical but can be rationalized. Monetary freedom, often associated with low inflation and stable currency, is generally seen as positive. However, if neighboring countries exhibit excessively high monetary freedom, it might lead to a relative appreciation of their currencies, making them more attractive for travel and investment at the expense of the domestic market. This phenomenon aligns with the substitution effect in economic theory,

where consumers and investors shift their preferences based on relative cost advantages. Similar findings are reported by Dreher [74], who observed that monetary stability in one country could sometimes divert tourism and investment flows from neighboring countries.

Finally, the specific impact of the Investment Freedom Index and Government Integrity Index on increasing domestic arrivals underscores the importance of a transparent and open investment climate. These indices signify the ease of investment flows and the reduction of corruption, essential for fostering investor confidence and sustainable economic growth. Conversely, the insignificant effects of the Financial Freedom Index, Tax Burden Index, and Government Spending Index suggest that these factors may not directly influence domestic economic activities or arrivals. This aligns with studies like Barro and Redlick [75], which indicate that while fiscal policies are crucial, their direct impact on short-term economic activities, such as tourism or immediate business decisions, might be limited compared to regulatory and governance factors.

## Conclusions and policy implications

### Conclusions

According to tourism-led growth models, economic freedom and its components are expected to play a significant role in the international tourism industry and boost tourism arrivals by generating externalities. Using these theories, the primary objective of this research was to examine the effects of economic freedom spillover from neighboring countries on tourist arrivals in European countries. Moreover, this study examined the spatial interaction between these countries' tourism industries and the spatial movement of economic freedom within the region. Diagnostic test analyses validated regional tourism growth's spatial characteristics and economic freedom's spillover effects on European tourism arrivals. This finding is consistent with Karimi et al. [18], who found evidence of regional spatial interaction in the tourism industry. The spatial Durbin model was chosen as the best spatial econometric model for studying the factors influencing regional tourism arrivals.

The analysis reveals that while certain model variables align with theoretical expectations, others highlight unique regional dynamics. The empirical evidence suggests that an increase in the GDP per capita of neighboring nations is associated with a rise in domestic tourist arrivals. This phenomenon can be attributed to the spillover effects of developed tourism sectors in neighboring countries on the home country's tourism industry. Such a correlation aligns with theoretical projections and resonates with the empirical findings of prior studies [18,36,76,77].

The underlying mechanism indicates that economic prosperity enhances a country's tourist infrastructure. Simultaneously, economic growth in neighboring countries boosts international demand for the home country's tourism offerings due to higher income levels in these neighboring nations. Moreover, as adjacent countries develop their tourism infrastructure through economic growth, they facilitate tourist mobility to regions with more sophisticated tourism amenities. Consequently, this interconnectivity increases regional tourist presence, further boosting tourist arrivals in the home country. This synergy between economic growth and tourism development underscores the complex interplay of regional economic dynamics and their impact on the tourism sector.

Additionally, the results indicate that an increase in neighboring countries' prices leads to an increase in inbound tourists, corroborated by Masiero & Nicolau [46] and Kang et al. [78]. Furthermore, trade liberalization significantly impacts tourism arrivals, which aligns with the findings of several empirical studies [79–83].

One of the paper's central findings is that the total economic freedom index has a meaningless and low-level positive significant effect on tourism arrivals in the European region. This

finding is consistent with that of Aslan et al. [15]. The European region is made up of numerous nations that enjoy varying degrees of economic freedom. Switzerland, for example, is one of the world's five truly "free" economies; Ukraine, Russia, and Belarus, among the other eight nations, are classified as repressive and primarily unfree. Poland, Italy, France, and Spain, among the other 17 nations, have moderately free economies, while only 18 of 45 European countries have economies classified as "mostly free". In addition, these countries are struggling with a variety of policy impediments, including costly labor regulations and excessively protective, various market-distorting subsidies, increased tax burdens, and persistent fiscal problems resulting from public-sector expansion. As a result, economic growth has been stagnant, exacerbating the burden of fiscal deficits and mounting debt in several countries throughout the region. Taken together, it is acceptable that the economic freedom index has a negligible effect on tourism arrivals in Europe. Furthermore, while the coefficients for the six economic freedom indices are statistically insignificant, the coefficients for the Trade Freedom Index, the Investment Freedom Index, the Property Rights Index, and the Government Integrity Index are statistically positive and significant. According to Heritage Foundation reports, these findings are significant because the European region's average scores on judicial effectiveness, property rights, investment freedom, financial freedom, and government integrity outperform the global averages.

Moreover, the majority of economic freedom indicators, including the Business Freedom, Trade Freedom, Labor Freedom, Investment Freedom, and Government Integrity indices, indicate that each country has a sizable positive influence, and only improvements in neighboring countries' Investment Freedom and Government Integrity measures will benefit the domestic tourism industry. According to Kırant Yozcu [84], the effect of each of these components confirms the previously mentioned positive relationship but does not consider the effect of these factors in neighboring countries. Furthermore, contrary to Aslan et al. [15], there is no evidence that the Financial Freedom, Government Spending, and Tax Burden indices significantly impact the tourism industry in the sample countries. This is unsurprising in light of our sample's financial development levels.

## Policy implications

The findings of our research underscore the significant role of economic freedom and government integrity in shaping inbound tourism, particularly within the European Union. Policymakers must prioritize enhancing economic freedom as a means to boost tourism. The relationship between economic freedom and inbound tourism is clear: higher levels of economic freedom correlate with increased tourist arrivals. This suggests that countries should adopt policies aimed at improving the Trade Freedom Index and Investment Freedom Index, which are vital for attracting international visitors.

Furthermore, the development of tourism infrastructure is essential for leveraging the benefits of economic freedom. Investments in transportation, hospitality, and local attractions not only enhance the tourist experience but also create a favorable environment for increased tourist flows. Policymakers should thus focus on developing robust tourism infrastructure that complements efforts to liberalize trade and encourage foreign investment.

Additionally, the spillover effects from economic freedom in neighboring countries illustrate the importance of regional cooperation. Enhancing tourism infrastructure in one country can lead to positive tourism spillovers for its neighbors. Therefore, it is crucial for policymakers to engage in coordinated strategies that recognize and capitalize on these interdependencies.

To maximize the potential for tourism growth, it is imperative to ensure that property rights are protected and that governance is transparent and accountable. These factors create

a business-friendly environment that attracts tourists and supports sustainable tourism development. By understanding the spatial interactions within the region's tourism sectors, policymakers can develop comprehensive strategies that not only enhance their own tourism industries but also contribute to regional tourism success.

## Limitation and future research

While this study provides valuable insights, it is important to acknowledge several limitations. First, the focus on European countries may limit the generalizability of the findings, particularly in regions with distinct economic, cultural, and political contexts. As such, the applicability of the results to non-European areas should be approached with caution. Moreover, the study's reliance on specific datasets may introduce data constraints that affect the robustness of our conclusions. Future research should consider employing a broader array of data sources to capture external factors that could influence the model's outcomes. Additionally, incorporating various development indicators, such as GDP per capita, could enhance our understanding of the model's applicability across different economic environments. To address these limitations, we recommend conducting sensitivity analyses in future studies, which could include examining data subsets or comparing countries within and outside the European Union. By taking these steps, future research can strengthen the validity of findings and provide a more comprehensive understanding of the complex dynamics at play.

## Supporting information

**S1 Dataset. Raw dataset.**
(XLSX)

## Author contributions

**Conceptualization:** Sakar Hasan Hamza.

**Data curation:** Mohsen Khezri.

**Formal analysis:** Sakar Hasan Hamza.

**Investigation:** Sakar Hasan Hamza, Qingna Li.

**Methodology:** Mohsen Khezri.

**Project administration:** Qingna Li.

**Resources:** Sakar Hasan Hamza.

**Software:** Mohsen Khezri.

**Supervision:** Qingna Li.

**Validation:** Sakar Hasan Hamza, Mohsen Khezri.

**Visualization:** Sakar Hasan Hamza.

**Writing – original draft:** Sakar Hasan Hamza, Qingna Li.

**Writing – review & editing:** Sakar Hasan Hamza, Mohsen Khezri.

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
