## [Decision Letter · Decision Letter 0]

17 Jul 2023

PONE-D-23-19081Economic Freedom Index Effects on Tourist’s Arrivals in European Countries: A Spatial Panel AnalysisPLOS ONE

Dear Dr. Hasan Hamza,

Thank you for submitting your manuscript to PLOS ONE. After careful consideration, we feel that it has merit but does not fully meet PLOS ONE’s publication criteria as it currently stands. Therefore, we invite you to submit a revised version of the manuscript that addresses the points raised during the review process.

Please revise this paper according to the reviewers comments. Please submit your revised manuscript by Aug 31 2023 11:59PM. If you will need more time than this to complete your revisions, please reply to this message or contact the journal office at plosone@plos.org . Please include the following items when submitting your revised manuscript:

We look forward to receiving your revised manuscript.

Kind regards,

Nishanthi Rupika Abeynayake, Ph.D

Academic Editor

PLOS ONE

3. Please amend your authorship list in your manuscript file to include author Sakar Hasan Hasan Hamza.

Reviewers' comments:

Reviewer's Responses to Questions

**Comments to the Author**

1. Is the manuscript technically sound, and do the data support the conclusions?

Reviewer #1: Yes

Reviewer #2: Partly

Reviewer #3: Partly

2. Has the statistical analysis been performed appropriately and rigorously? 

Reviewer #1: Yes

Reviewer #2: Yes

Reviewer #3: Yes

3. Have the authors made all data underlying the findings in their manuscript fully available?

Reviewer #1: Yes

Reviewer #2: Yes

Reviewer #3: Yes

4. Is the manuscript presented in an intelligible fashion and written in standard English?

Reviewer #1: Yes

Reviewer #2: Yes

Reviewer #3: Yes

5. Review Comments to the Author

Reviewer #1: Authors have given considerable attention to the analysis of the data. However, it needs a better a structure. Writing style and gramma must be revised. Please refer to the comments made on manuscript.

Reviewer #2: I have thoroughly reviewed your paper and would like to provide some constructive feedback to enhance its effectiveness. It would be beneficial to place greater emphasis on the crucial role of tourism at the global level, while also highlighting the interconnection between tourism and economic freedom. By incorporating these aspects into your introduction, you can create a more comprehensive and professional piece of writing.

While you have provided a good starting point, I would suggest focusing on specific theories that are particularly relevant to the topic. The following theories could be incorporated into your analysis: Tourism-led Economic Growth Theory, Input-Output Analysis, Tourism Satellite Account (TSA), Destination Competitiveness Theory, Economic Multiplier Theory, and Social Exchange Theory. These theories offer robust frameworks for understanding the economic impacts and benefits associated with the tourism industry.

Furthermore, it is important to note that while these theories provide valuable insights, they are not exhaustive, and the selection of theories should be tailored to the specific context or region being examined. It would be beneficial to support your arguments by referring to empirical research, statistical analysis, and relevant case studies. This approach will enhance the validity and credibility of your findings, allowing you to effectively demonstrate the critical role of the tourism industry in driving economic prosperity.

The literature review appears to lack a clear logical flow, making it difficult for readers to follow the progression of ideas. Consider organizing the review in a more structured manner, such as by thematic categories or chronological order, to enhance its coherence and readability.

It would be valuable to synthesize the findings and conclusions from the reviewed literature. Highlight the key themes, trends, and conflicting perspectives that emerge from the existing studies. This synthesis will not only provide a comprehensive overview but also help establish the rationale for your research and its contribution to the field.

Ensure that your citations are consistently and accurately formatted throughout the literature review. Properly attributing the sources will enhance the credibility and professionalism of your work.

To improve clarity, consider breaking down complex equations into smaller components and clearly defining each parameter used. Additionally, provide explanations or contextual information to help readers understand the purpose and meaning of the equations in the context of your study. Ensure that all parameters used in your methodology are described in a logical and easily understandable manner. Avoid assuming prior knowledge on the part of the reader. Provide clear explanations for each parameter, including their definitions.

Reviewer #3: Title:

The title of the article is both well-chosen and appropriate for the content it encompasses. It effectively captures the essence of the research and provides readers with a clear understanding of the article's focus.

Abstract & keywords:

The abstract is poorly structured. Highlights of problem identification need to be included in the abstract. Highlighting the problem identification and stating the objectives of the study clearly with aid to understand the originality of the work.

Moreover, certain claims on the economic freedom index seemed to be vague.

Examples:

1. More importantly, the economic freedom index has a significant effect on tourism arrivals in Europe

• Need to clarify the nature and direction of the impacts of the index

2. While some components of economic freedom have marginal effects,

• This is again a vague statement and the author needs to be more specific in reporting his results even in the abstract

3. Our findings have meaningful implications for policymakers interested in sustainable tourism in European countries

• Better to state what these implications are in brief

Better to rewrite the abstract considering the above points.

Keywords: well-chosen and appropriate

Introduction:

The introduction sets out a good platform for the study.

However, the identification of the research problem is not well justified.

Certain terms need to be defined in order to facilitate the understanding (ex: four significant categories of economic freedom by Heritage Foundation need to be defined. the rule of law, market openness, regulatory efficiency, and government size)

Better to expand the references rather than listing them all together in one place, so that the reader can understand which author has found out which relationship. (ex: few studies have examined the effects of press and personal freedom, civil liberties, and economic and political freedom, corruption, and economic freedom (Su and Lin 2014; Kubickova 2016; Saha, Su, and Campbell 2017; Tang 2018; Kubickova 2019; Bulut, Kocak, and Suess 2020; Öcal and Yildirim 2010))

Literature review:

The author shows a good understanding of broad, relevant literature. The review provides a comprehensive overview of different indices and their impacts on tourism.

A few issues are being identified in relation to the formatting and presentation of the section.

Certain citation formats need to be revised (Ex: first reference in the literature review section, Saha and Yap ((2014) on page no 05).

Certain references are too old to be used and highlight the need to use recent articles.

Methodology:

Methodology and the relevant models are well-explained and suitable for the study.

Results & Discussion:

This section provides very good analyses and clarifications to support answering the research questions.

Author had made a commendable attempt to relate the findings of the study to the previous literature.

Conclusions:

Better to include a separate section for conclusions, implications and future research.

References:

Referencing is satisfactory. However, need to revisit to the section as certain references cited in the text are not included in the list of references (ex: Heritage Foundation 2020) on page number 2 second paragraph).

Language and formatting

The quality of English and general presentation is satisfactory.

While the expression is generally clear, there are a few issues in general word formatting, punctuation and grammar.

6. PLOS authors have the option to publish the peer review history of their article (what does this mean? ). If published, this will include your full peer review and any attached files.

**Do you want your identity to be public for this peer review?** For information about this choice, including consent withdrawal, please see our Privacy Policy .

Reviewer #1: No

Reviewer #2: **Yes: ** TPSR Guruge

Reviewer #3: No

---

## [Author Response · Author response to Decision Letter 1]

14 Oct 2023

Please see attached file.

Manuscript PONE-D-23-19081

Title: Economic Freedom Index Effects on Tourist’s Arrivals in European Countries: A Spatial Panel Analysis

Journal: PLOS ONE

Dear Editor and Reviewers,

Thank you very much for the opportunity to revise and resubmit our manuscript. We appreciate the constructive and careful review of our manuscript by the reviewer and editor. We have attempted to address all issues raised in the review, either in the text itself and/or in our comments. We truly believe that the manuscript is in much better shape as a result of the changes we made in response to your critical comments. Of course, we are happy to revisit any of the issues raised by the reviewers and make additional changes to improve the overall quality of the work.

Below, we provide a point-by-point response to each of the concerns raised by the reviewer.

Reviewer’s comments:

Reviewer #1:

Authors have given considerable attention to the analysis of the data. However, it needs a better a structure. Writing style and gramma must be revised. Please refer to the comments made on manuscript.

Author: Thank you very much for your comment. In the revised version of the manuscript, we have attempted to address all these issues.

……………………………………………

Reviewer #2:

1. It would be beneficial to place greater emphasis on the crucial role of tourism at the global level, while also highlighting the interconnection between tourism and economic freedom. By incorporating these aspects into your introduction, you can create a more comprehensive and professional piece of writing.

Author: Thank you for your comment. The introduction has been rewritten. The highlighted sections have been added to the introduction according to the comments. Other text has been rewritten for better clarity.

2. While you have provided a good starting point, I would suggest focusing on specific theories that are particularly relevant to the topic. The following theories could be incorporated into your analysis: Tourism-led Economic Growth Theory, Input-Output Analysis, Tourism Satellite Account (TSA), Destination Competitiveness Theory, Economic Multiplier Theory, and Social Exchange Theory. These theories offer robust frameworks for understanding the economic impacts and benefits associated with the tourism industry. Furthermore, it is important to note that while these theories provide valuable insights, they are not exhaustive, and the selection of theories should be tailored to the specific context or region being examined. It would be beneficial to support your arguments by referring to empirical research, statistical analysis, and relevant case studies. This approach will enhance the validity and credibility of your findings, allowing you to effectively demonstrate the critical role of the tourism industry in driving economic prosperity.

Author: By examining all the theories proposed, we found the Destination Competitiveness Theory suitable for the purposes of the article. This theory is used in the manuscript.

3. The literature review appears to lack a clear logical flow, making it difficult for readers to follow the progression of ideas. Consider organizing the review in a more structured manner, such as by thematic categories or chronological order, to enhance its coherence and readability. It would be valuable to synthesize the findings and conclusions from the reviewed literature. Highlight the key themes, trends, and conflicting perspectives that emerge from the existing studies. This synthesis will not only provide a comprehensive overview but also help establish the rationale for your research and its contribution to the field.

Author: According to the points raised, we have made significant changes in the literature review section. We tried to apply the comments, especially the last paragraph has been added to highlight the key themes, trends, and conflicting perspectives.

4. Citations are consistently and accurately formatted throughout the literature review. Properly attributing the sources will enhance the credibility and professionalism of your work.

Author: All citations were rechecked and converted to journal format with Mendeley software.

5. To improve clarity, consider breaking down complex equations into smaller components and clearly defining each parameter used. Additionally, provide explanations or contextual information to help readers understand the purpose and meaning of the equations in the context of your study. Ensure that all parameters used in your methodology are described in a logical and easily understandable manner. Avoid assuming prior knowledge on the part of the reader. Provide clear explanations for each parameter, including their definitions.

Author: Explaining the methodology and modeling of spatial panel models is complex and extensive, and breaking and explaining the equations will lead to the expression of newer equations that need to be interpreted again. We have referenced studies related to the article's methodology for a more complete explanation. More important than understanding the equations is the interpretation of the estimated parameters of the variables, which is no different from other conventional panel models. However, if our explanation is still not convincing, we will try to delineate this more should we get another round of revisions.

Reviewer #3:

1. Abstract & keywords:

The abstract is poorly structured. Highlights of problem identification need to be included in the abstract. Highlighting the problem identification and stating the objectives of the study clearly with aid to understand the originality of the work.

Moreover, certain claims on the economic freedom index seemed to be vague.

Examples:

1. More importantly, the economic freedom index has a significant effect on tourism arrivals in Europe

• Need to clarify the nature and direction of the impacts of the index

2. While some components of economic freedom have marginal effects,

• This is again a vague statement and the author needs to be more specific in reporting his results even in the abstract

3. Our findings have meaningful implications for policymakers interested in sustainable tourism in European countries

• Better to state what these implications are in brief

Better to rewrite the abstract considering the above points.

Author: Thank you very much for your detailed comment. The abstract has been rewritten according to the points raised.

2. Introduction:

The introduction sets out a good platform for the study.

However, the identification of the research problem is not well justified.

Certain terms need to be defined in order to facilitate the understanding (ex: four significant categories of economic freedom by Heritage Foundation need to be defined. the rule of law, market openness, regulatory efficiency, and government size)

Better to expand the references rather than listing them all together in one place, so that the reader can understand which author has found out which relationship. (ex: few studies have examined the effects of press and personal freedom, civil liberties, and economic and political freedom, corruption, and economic freedom (Su and Lin 2014; Kubickova 2016; Saha, Su, and Campbell 2017; Tang 2018; Kubickova 2019; Bulut, Kocak, and Suess 2020; Öcal and Yildirim 2010)).

Author: In line with the comments raised by Reviewer 2 and these comments, the introduction has been rewritten considering. The highlighted sections have been added to the introduction according to the comments. Other text has been rewritten for better clarity.

3. Literature review:

A few issues are being identified in relation to the formatting and presentation of the section.

Certain citation formats need to be revised (Ex: first reference in the literature review section, Saha and Yap ((2014) on page no 05).

Author: All citations were rechecked and converted to journal format with Mendeley software.

4. Certain references are too old to be used and highlight the need to use recent articles.

Author: According to the points raised, the entire literature review section of the research was rewritten and some new articles are added. We tried to apply the comments, especially the last paragraph has been added to highlight the key themes, trends, and conflicting perspectives.

5. Conclusions:

Better to include a separate section for conclusions, implications and future research.

Author: this section is divided into the conclusions and implications sections.

6. References:

Referencing is satisfactory. However, need to revisit to the section as certain references cited in the text are not included in the list of references (ex: Heritage Foundation 2020) on page number 2 second paragraph).

Author: All citations were rechecked and converted to journal format with Mendeley software.

7. Language and formatting

The quality of English and general presentation is satisfactory.

While the expression is generally clear, there are a few issues in general word formatting, punctuation and grammar.

Author: The introduction and the background of the research were rewritten to fix possible writing errors.

---

## [Decision Letter · Decision Letter 1]

24 Jan 2024

PONE-D-23-19081R1Economic Freedom Index Effects on Inbound Tourism in European Countries: A Spatial AnalysiPLOS ONE

Dear Dr. Hamza,

Thank you for submitting your manuscript to PLOS ONE. After careful consideration, we feel that it has merit but does not fully meet PLOS ONE’s publication criteria as it currently stands. Therefore, we invite you to submit a revised version of the manuscript that addresses the points raised during the review process.

We look forward to receiving your revised manuscript.

Kind regards,

Thang Quyet Nguyen, Ph.D

Academic Editor

PLOS ONE

Additional Editor Comments:

The revised verson has been improved much. The manuscript needs to be supplemented with theoretical foundations. The methods of analysis and techniques used need to be further clarified. It is particularly important for the manuscript needs to clearly highlight the newly discovered points from the research.

Reviewers' comments:

Reviewer's Responses to Questions

**Comments to the Author**

1. If the authors have adequately addressed your comments raised in a previous round of review and you feel that this manuscript is now acceptable for publication, you may indicate that here to bypass the “Comments to the Author” section, enter your conflict of interest statement in the “Confidential to Editor” section, and submit your "Accept" recommendation.

Reviewer #1: All comments have been addressed

Reviewer #4: (No Response)

Reviewer #5: (No Response)

Reviewer #6: (No Response)

2. Is the manuscript technically sound, and do the data support the conclusions?

Reviewer #1: Yes

Reviewer #4: Partly

Reviewer #5: (No Response)

Reviewer #6: Yes

3. Has the statistical analysis been performed appropriately and rigorously? 

Reviewer #1: Yes

Reviewer #4: N/A

Reviewer #5: (No Response)

Reviewer #6: Yes

4. Have the authors made all data underlying the findings in their manuscript fully available?

Reviewer #1: Yes

Reviewer #4: No

Reviewer #5: (No Response)

Reviewer #6: Yes

5. Is the manuscript presented in an intelligible fashion and written in standard English?

Reviewer #1: Yes

Reviewer #4: No

Reviewer #5: (No Response)

Reviewer #6: Yes

6. Review Comments to the Author

Reviewer #1: Author/s have addressed all comments satisfactorily. However, they could consider reducing the number of unmercenary tables and equations as it breaks the flow. Measurement of variables can be showed using equations.

Reviewer #4: To Editor:

The authors have made considerable attempts to address issues raised in the previous reviews. However, the present revision still needs further attention, including the methodology and result sections for a better structure. The detailed comments as below.

Abstract

The abstract of an empirical study should be more specific by reporting results in numbers rather than merely qualitative statements, as in the current revision; brief policy implications need to be included in this applied research and in specific contexts.

Methodology

- The choice of Destination Completeness Theory (DCT) should be further justified to replace others, such as the Tourism-led Growth Theory which seems to be more relevant to the topic and has already been mentioned in the literature review section. The introduction of DCT should be put in the review section to make a more logical flow to build a framework.

- Consider organizing the methodology section to delve mainly into a framework or theoretical hypothesis (if any) that emerged from previous sections and provide explanations or justifications for the choice of the empirical strategy or specification of regression equations. This elaboration not only establishes the rationale for the research idea but also enhances the robustness of the estimated results.

Results/Data

- The database does not fully describe whether tourists come from only the EU or other countries. It should be from only the EU. A description of the countries list should also be added (maybe in the appendix) as the present data only code numbers instead of names of countries.

- The concepts and terms of economic freedom and its components need to be clearly defined by the authors rather than letting the reader guess, such as the Trade Openness. This can impact the study's outcomes since there are several different measures of trade openness (e.g., export, import, or the total of both to GDP).

- The constructed variables and their descriptive statistics should be put in the methodology section together with the data section or in a separate section for better paper structure and to avoid disfunctioning the results section, which focuses on only the estimated results or main outcomes of the study.

/Estimation of the results

- This section causes much confusion and lacks consistency between paragraphs second and third, and within each one there is also inconsistency among sentences, mainly because of not indicating the terms home country and neighboring countries in many contexts. E.g., some of them are: “the increase in destination country prices positively affects tourist arrivals”; “As per theoretical predictions, an increase in the price level should increase the prices of goods and services in the destination countries, thereby enhancing the destination country's competitiveness”; “A review of Figure 2 reveals that for 170 countries in 2017, the relationship between price levels and tourist arrivals is a positive communication pattern.” “Even an increase in neighboring countries' prices results in a flow of tourists to neighboring countries and a decline in entries.”

- Since the data is not described clearly, as mentioned above, this statement needs to be fully justified and more specific: “The increase could be attributed to the arrival of tourists from neighboring countries, regional tourism industry development, or the overall increase in tourists in the developed world.”

- The authors should use the usual way of presenting p-values for easier reading, such as *, **, and *** for different levels of significant statistics; several tables should be put in the appendix to spare space for other main results with further discussion.

Because the specification of regression models and their choices of variables are often subjective and there is a complicated nexus between each country’s price level and tourist arrivals, the authors should provide a separate section or some validity tests and analyses on the robustness of the estimated results. Several sensitive tests may include using different subsets of the data (e.g., the European countries with and without the European Union); different development levels (e.g., GDP per capita); etc.

Others

Some grammatical and typing/formating errors remain, such as: “under researched,” “comprehensive comprehension,” “Dependency Critique Theory (DCT),” “data,” “Estimation of result.” The whole paper should be rechecked for consistency in grammar, formatting, and structure.

Reviewer #5: Manuscript Number: PONE-D-23-19081R1

Title: Economic Freedom Index Effects on Inbound Tourism in European Countries: A Spatial Analysis

1. Abstract & keywords:

Even though the second verson has been improved, it is better for the authors to emphasize on the significance or the contribution of this paper. At the moment, authors seems to spend too much on writing the results.

2. Introduction:

The revised verson has been improved much. However, there are still some minor issues. First of all, the brief introduction of the key issues are still missing. Authors should include the defintion in the first or second paragraph. Secondly, the research gap has been poorly emphasized in the introduction. Finally, authors might consider including the research objectives at the end of the introduction.

3. Literature review

While authors have included several studies, a logical focus is still missing. Authors should try to emphasize on the key point of each section, instead of just listing study.

4.Methodology

The structure of this section is quite strange.

The introduction of Destination Competitiveness Theory should be placed in the introduction or Literature review, not in the methodology. Second, objectives should be in the Introduction

Methodology should focus on how to collect and analyze the data.

5. Results:

6. Discussion

In the conclusion section, authors should not dicuss the data results. It should be in the Results section. Dicussion should focus on the findings of the study and its contribution.

There shoul be one separate section about limitation and future research (extend the last 3 sentences)

7. Language and formatting

While the language quality is acceptable, there are still minors grammar mistake. Please doublcheck.

Reviewer #6: Dear Author

I have read your previous notes and corrections. The changes you made are quite good and make your paper more interesting. However, I feel there is still something you need to complete, namely there needs to be a discussion section. This is considered very necessary because in the discussion section you will discuss reflections on all your findings, apart from that you can also argue the value of objectivity between theory and findings. Thank you

7. PLOS authors have the option to publish the peer review history of their article (what does this mean? ). If published, this will include your full peer review and any attached files.

**Do you want your identity to be public for this peer review?** For information about this choice, including consent withdrawal, please see our Privacy Policy .

Reviewer #1: No

Reviewer #4: No

Reviewer #5: No

Reviewer #6: **Yes: ** Yandra Rivaldo

---

## [Author Response · Author response to Decision Letter 2]

17 Feb 2024

Authors' responses to comments from the Editor and the Reviewer

Manuscript PONE-D-23-19081

Title: Economic Freedom Index Effects on Tourist's Arrivals in European Countries: A Spatial Panel Analysis

Journal: PLOS ONE

Authors:

We would like to thank the editor and the reviewer for taking time to carefully read the manuscript and providing very useful comments. We have accounted for all the changes suggested to the best of our abilities. Your insightful comments and suggestions have improved the quality of the manuscript significantly. We hope the revision is to your satisfaction.

Reviewer #4:

1- Abstract

The abstract of an empirical study should be more specific by reporting results in numbers rather than merely qualitative statements, as in the current revision; brief policy implications need to be included in this applied research and in specific contexts.

Authors: According to the reviewer's opinion, the Introduction was rewritten.

2- Methodology

- The choice of Destination Completeness Theory (DCT) should be further justified to replace others, such as the Tourism-led Growth Theory which seems to be more relevant to the topic and has already been mentioned in the literature review section. The Introduction of DCT should be put in the review section to make a more logical flow to build a framework.

- Consider organizing the methodology section to delve mainly into a framework or theoretical hypothesis (if any) that emerged from previous sections and provide explanations or justifications for the choice of the empirical strategy or specification of regression equations. This elaboration not only establishes the rationale for the research idea but also enhances the robustness of the estimated results.

Authors: According to the reviewer's recommendation, we investigated the Tourism-led Growth Theory. This theory talks about the effects of tourism on economic growth (while our study examines the effect of economic growth on tourism). Therefore, it is not compatible with the theoretical principles of our study. Due to the objections of other reviewers, we had to summarize this section to preserve the methodology section's comprehensiveness. Therefore, we transferred the initial materials to the literature review section.

3- Results/Data

- The database does not fully describe whether tourists come from only the EU or other countries. It should be from only the EU. A description of the countries list should also be added (maybe in the appendix) as the present data only code numbers instead of names of countries.

Authors: As highlighted in Table 1, total tourists arrival have been examined in this study. We put the list of countries on page 9 as a footnote according to the opinion of the honorable reviewer.

- The concepts and terms of economic freedom and its components need to be clearly defined by the authors rather than letting the reader guess, such as the Trade Openness. This can impact the study's outcomes since there are several different measures of trade openness (e.g., export, import, or the total of both to GDP).

Authors: Necessary corrections were made in Table 1. It should be noted that the indicators of economic freedom are explained after Table 2.

- The constructed variables and their descriptive statistics should be put in the methodology section together with the data section or in a separate section for better paper structure and to avoid disfunctioning the results section, which focuses on only the estimated results or main outcomes of the study.

Authors: We have modified the structure of the article to apply the reviewer's opinion

4- Estimation of the results

- This section causes much confusion and lacks consistency between paragraphs second and third, and within each one there is also inconsistency among sentences, mainly because of not indicating the terms home country and neighboring countries in many contexts. E.g., some of them are: "the increase in destination country prices positively affects tourist arrivals"; "As per theoretical predictions, an increase in the price level should increase the prices of goods and services in the destination countries, thereby enhancing the destination country's competitiveness"; "A review of Figure 2 reveals that for 170 countries in 2017, the relationship between price levels and tourist arrivals is a positive communication pattern." "Even an increase in neighboring countries' prices results in a flow of tourists to neighboring countries and a decline in entries."

- Since the data is not described clearly, as mentioned above, this statement needs to be fully justified and more specific: "The increase could be attributed to the arrival of tourists from neighboring countries, regional tourism industry development, or the overall increase in tourists in the developed world."

Authors: We read all the mentioned sections and other sections. The text has been completely rewritten and revised to remove any ambiguities. Please refer to pages 12, 15 and 16.

- The authors should use the usual way of presenting p-values for easier reading, such as *, **, and *** for different levels of significant statistics; several tables should be put in the appendix to spare space for other main results with further discussion.

Authors: A number of tables were moved to the appendix and asterisks were placed.

- Because the specification of regression models and their choices of variables are often subjective and there is a complicated nexus between each country's price level and tourist arrivals, the authors should provide a separate section or some validity tests and analyses on the robustness of the estimated results. Several sensitive tests may include using different subsets of the data (e.g., the European countries with and without the European Union); different development levels (e.g., GDP per capita); etc.

Author: A primary concern highlighted by fellow reviewers pertains to the extensive inclusion of computational tables within the article. Specifically, Tables 3 through 6 meticulously assess the array of diagnostic tests for optimal model selection prevalent in empirical studies. Furthermore, given the spatial nature of our panel data model, it entails a considerable quantity of estimated model coefficients. This complexity is compounded as the incorporation of additional variables inversely affects the model's degrees of freedom, introducing limitations to our analysis. Despite these methodological challenges, we have duly acknowledged the esteemed referee's insights in the concluding section, proposing them as avenues for future research endeavors.

5- Others

Some grammatical and typing/formating errors remain, such as: "under researched," "comprehensive comprehension," "Dependency Critique Theory (DCT)," "data," "Estimation of result." The whole paper should be rechecked for consistency in grammar, formatting, and structure.

Authors: We fixed these problems. Then, the whole text is checked by an editor. Modifications can be seen in the form of change.

Reviewer #5:

1. Abstract & keywords:

Even though the second verson has been improved, it is better for the authors to emphasize on the significance or the contribution of this paper. At the moment, authors seems to spend too much on writing the results.

Authors: According to the reviewers' opinions, the Introduction was rewritten.

2. Introduction:

The revised verson has been improved much. However, there are still some minor issues. First of all, the brief Introduction of the key issues are still missing. Authors should include the defintion in the first or second paragraph. Secondly, the research gap has been poorly emphasized in the Introduction. Finally, authors might consider including the research objectives at the end of the Introduction.

Authors: According to the reviewer's opinion, the Introduction was restructured entirely, and requested sections were added.

3. Literature review

While authors have included several studies, a logical focus is still missing. Authors should try to emphasize on the key point of each section, instead of just listing study.

Authors: We have summarized the key points at the end of this section.

4.Methodology

The structure of this section is quite strange.

The Introduction of Destination Competitiveness Theory should be placed in the Introduction or Literature review, not in the methodology. Second, objectives should be in the Introduction

Methodology should focus on how to collect and analyze the data.

Authors: we had to summarize this section to preserve the methodology section's comprehensiveness. Therefore, we transferred the initial materials to the literature review section.

6. Discussion

In the conclusion section, authors should not dicuss the data results. It should be in the Results section. Dicussion should focus on the findings of the study and its contribution.

There shoul be one separate section about limitation and future research (extend the last 3 sentences)

Authors: This problem has been fixed in the modified version, and future research is extended in the last paragraph.

7. Language and formatting

While the language quality is acceptable, there are still minors grammar mistake. Please doublcheck.

Authors: We fixed these problems. Then, the whole text is checked by an editor. Modifications can be seen in the form of change.

Reviewer #6:

I have read your previous notes and corrections. The changes you made are quite good and make your paper more interesting. However, I feel there is still something you need to complete, namely there needs to be a discussion section. This is considered very necessary because in the discussion section you will discuss reflections on all your findings, apart from that you can also argue the value of objectivity between theory and findings. Thank you.

Authors: Thanks for the reviewer's opinion. We have made some corrections in the conclusion section. We have considered this structure due to the compulsion to provide the opinion of all the honorable reviewers.

---

## [Editor Report · Decision Letter 2]

5 Mar 2024

PONE-D-23-19081R2Economic Freedom Index Effects on Inbound Tourism in European Countries: A Spatial AnalysiPLOS ONE

Dear Dr. Hamza,

Thank you for submitting your manuscript to PLOS ONE. After careful consideration, we feel that it has merit but does not fully meet PLOS ONE’s publication criteria as it currently stands. Therefore, we invite you to submit a revised version of the manuscript that addresses the points raised during the review process.

**ACADEMIC EDITOR: **The manuscript needs to be meticulously and carefully edited. It should provide a more comprehensive review of the literature and updated information. The methods of analysis and techniques used need to be further clarified. Importantly, the manuscript should prominently highlight the new findings discovered through the research. It is particularly important for the article needs to clearly highlight the newly discovered points from the research. The format and language used are also important areas of concern for the article.

I am also enclosing the comprehensive comments from reviewers here for you to make necessary improvements to the article 

We look forward to receiving your revised manuscript.

Kind regards,

Thang Quyet Nguyen, Ph.D

Academic Editor

PLOS ONE

---

## [Author Response · Author response to Decision Letter 3]

22 Mar 2024

Author: Manuscript Number: PONE-D-23-19081R1

Title: Economic Freedom Index Effects on Inbound Tourism in European Countries: A Spatial Analysis

1. Abstract & keywords:

Even though the second verson has been improved, it is better for the authors to emphasize on the significance or the contribution of this paper. At the moment, authors seems to spend too much on writing the results.

Author: The abstract has been rewritten with more emphasis on the contribution of the article.

2. Introduction:

The revised verson has been improved much. However, there are still some minor issues. First of all, the brief introduction of the key issues are still missing:

Author: The defects are completely fixed. Please see below for the exact address of each fixed problem.

Authors should include the defintion in the first or second paragraph.

Author: The problem has been fixed. Page 1, first paragraph.

Secondly, the research gap has been poorly emphasized in the introduction.

Author: The problem has been fixed. Page 3, last paragraph.

Finally, authors might consider including the research objectives at the end of the introduction.

Author: The problem has been fixed. Page 4, first paragraph.

3. Literature review

While authors have included several studies, a logical focus is still missing. Authors should try to emphasize on the key point of each section, instead of just listing study.

Author: The research literature was completely rewritten. We categorized the content, emphasizing on the key point of each section.

4.Methodology

The structure of this section is quite strange.

The introduction of Destination Competitiveness Theory should be placed in the introduction or Literature review, not in the methodology. Second, objectives should be in the Introduction

Methodology should focus on how to collect and analyze the data.

Author: Irrelevant content was removed and defects were fixed. Please refer to the end of page 8.

5. Results:

6. Discussion

In the conclusion section, authors should not dicuss the data results. It should be in the Results section. Dicussion should focus on the findings of the study and its contribution.

There shoul be one separate section about limitation and future research (extend the last 3 sentences)

Author: We have carefully reviewed this section and removed any material related to data analysis. Also, the research limitations section was added at the end along with the expanded recommendations. Please refer to page 18, last paragraph.

7. Language and formatting

While the language quality is acceptable, there are still minors grammar mistake. Please doublcheck.

Author: We re-read the article and fixed every writing problem that we found.

---

## [Decision Letter · Decision Letter 3]

23 Jun 2024

PONE-D-23-19081R3Economic Freedom Index Effects on Inbound Tourism in European Countries: A Spatial AnalysiPLOS ONE

Dear Dr. Hamza,

Thank you for submitting your manuscript to PLOS ONE. After careful consideration, we feel that it has merit but does not fully meet PLOS ONE’s publication criteria as it currently stands. Therefore, we invite you to submit a revised version of the manuscript that addresses the points raised during the review process.

**ACADEMIC EDITOR: **I am providing the extensive feedback from reviewers for your consideration in order to enhance the manuscript. I emphasize that the explanation of the editing of the manuscript must be careful and detailed

We look forward to receiving your revised manuscript.

Kind regards,

Thang Quyet Nguyen, Ph.D

Academic Editor

PLOS ONE

Journal Requirements:

Reviewers' comments:

Reviewer's Responses to Questions

**Comments to the Author**

1. If the authors have adequately addressed your comments raised in a previous round of review and you feel that this manuscript is now acceptable for publication, you may indicate that here to bypass the “Comments to the Author” section, enter your conflict of interest statement in the “Confidential to Editor” section, and submit your "Accept" recommendation.

Reviewer #7: All comments have been addressed

Reviewer #8: (No Response)

2. Is the manuscript technically sound, and do the data support the conclusions?

Reviewer #7: No

Reviewer #8: Yes

3. Has the statistical analysis been performed appropriately and rigorously? 

Reviewer #7: No

Reviewer #8: Yes

4. Have the authors made all data underlying the findings in their manuscript fully available?

Reviewer #7: No

Reviewer #8: No

5. Is the manuscript presented in an intelligible fashion and written in standard English?

Reviewer #7: No

Reviewer #8: Yes

6. Review Comments to the Author

Reviewer #7: - The written introduction needs to clearly show the practical context: Need more data..., lack of theoretical context

- Apply the theoretical base

- Old references need updating

- The qualitative research part needs to be described in more detail such as: discussing the scale...

- Tables need deeper analysis.

- How should we compare the differences with other studies?

- The implication is that the writing needs to be more focused.

Reviewer #8: 1. I could not agree with your assertion that this is the first study that harnesses on a spatial analysis to unravel the relationship between economic freedom and tourism. For example, of existing studies in that direction see below:

Ouchen, A., & Montargot, N. (2022). Non-spatial and spatial econometric analysis of tourism demand in a panel of countries around the world. Spatial Economic Analysis, 17(2), 262-283.

Xie, W., Li, H., & Yin, Y. (2021). Research on the spatial structure of the European union’s tourism economy and its effects. International Journal of Environmental Research and Public Health, 18(4), 1389.

Based on this evidence, can you rephrase your statement and enhance your contribution.

2. The motivation of the study is not substantially justified given that your primary contribution is not grounded as there are numerous studies that have used the spatial approach, I am not convinced whether you have strongly justified your methodology and scope of study. Provide more information on the methods used by the existing studies in your literature review and tease out the gap you intend to fill. Since you failed to provide discernible evidence of the methodologies used in previous studies, it is difficult to accept that you are the first authors to use the methodology in your study.

3. Reconcile some statements in your conclusion and findings. For example;

“Furthermore, while the coefficients for six economic freedom indices are meaningless, the coefficients for the Trade Freedom Index, the Investment Freedom Index, the Property Rights Index, and the Government Integrity Index are significantly positive.” What does this statement mean? Insignificant or significant?

4. Can you justify why these findings are apparent? Provide some evidence-based rationale for these findings juxtaposing with previous studies.

“The neighboring countries' total economic freedom index demonstrates positive and significant effects. Furthermore, most economic freedom variables, such as the Business Freedom Index, the Investment Freedom Index, the Labor Freedom Index, the Trade Freedom Index, and the Government Integrity Index, exhibit a statistically significant and positive effect. However, neighboring countries' Monetary Freedom Index results in a decrease in the number of domestic arrivals. Among the various indicators, only improvements in the country's and neighboring countries' Investment Freedom Index and Government Integrity Index increase domestic arrivals. However, the Financial Freedom Index, the Tax Burden Index, and the Government Spending Index demonstrate no significant effect.”

7. PLOS authors have the option to publish the peer review history of their article (what does this mean? ). If published, this will include your full peer review and any attached files.

**Do you want your identity to be public for this peer review?** For information about this choice, including consent withdrawal, please see our Privacy Policy .

Reviewer #7: No

Reviewer #8: **Yes: **

---

## [Author Response · Author response to Decision Letter 4]

25 Jul 2024

Reviewer #7:

- The written introduction needs to clearly show the practical context: Need more data..., lack of theoretical context

- Apply the theoretical base

Author: In order to solve the above problem, we added the theoretical base section titled Destination Competitiveness Theory (DCT) to the study. Please refer to page 7.

Also, the contribution section of the article was modified to have a more practical and clear approach. Please refer to page 4.

- Old references need updating

Author: A number of new studies were also added to the article.

- The qualitative research part needs to be described in more detail such as: discussing the scale...

- Tables need deeper analysis.

Author: The estimation results section was rewritten for better clarity. See pages 12 and 13 for details.

- How should we compare the differences with other studies?

Author: The results of the study were compared with previous articles. Please refer to page 17 of the last paragraph.

- The implication is that the writing needs to be more focused.

Author: The Policy Implications results section was rewritten. Please refer to page 18 of the last paragraph and page 19 of the first paragraph.

Reviewer #8:

1. I could not agree with your assertion that this is the first study that harnesses on a spatial analysis to unravel the relationship between economic freedom and tourism. For example, of existing studies in that direction see below:

Ouchen, A., & Montargot, N. (2022). Non-spatial and spatial econometric analysis of tourism demand in a panel of countries around the world. Spatial Economic Analysis, 17(2), 262-283.

Xie, W., Li, H., & Yin, Y. (2021). Research on the spatial structure of the European union’s tourism economy and its effects. International Journal of Environmental Research and Public Health, 18(4), 1389.

Based on this evidence, can you rephrase your statement and enhance your contribution.

2. The motivation of the study is not substantially justified given that your primary contribution is not grounded as there are numerous studies that have used the spatial approach, I am not convinced whether you have strongly justified your methodology and scope of study. Provide more information on the methods used by the existing studies in your literature review and tease out the gap you intend to fill. Since you failed to provide discernible evidence of the methodologies used in previous studies, it is difficult to accept that you are the first authors to use the methodology in your study.

Author: Thank you for your comments. Of the two articles presented, only Ouchen and Montargot (2022) employed a spatial econometric approach. However, their model is incomplete and lacks the crucial table of marginal effects, which we have included as Table 9 in our article. This table is not generated by software like Stata; we had to use MATLAB programming to produce it. As stated in the contribution section of our article: "In essence, the contribution of this paper manifests across two principal domains: firstly, it comprehensively delves into the influence of economic freedom on the tourism industry, encompassing a broad array of economic freedom index components. Secondly, this study pioneers the adoption of a spatial panel data approach for examining the repercussions of economic freedom on tourism”, we did not claim that the spatial econometric method was used solely in our study. This article's primary innovation lies in investigating the effects of economic freedom indicators on the tourism industry. Even if we had conducted this study using other conventional econometric methods, it would represent a novel study. Obviously, we have made mistakes in clearly expressing the concepts, so we have tried to express the innovation of the study more clearly by referring to the studies mentioned by the honorable referee and other available studies to show that this methodology is used before. Please refer to page 4, first paragraph.

3. Reconcile some statements in your conclusion and findings. For example;

“Furthermore, while the coefficients for six economic freedom indices are meaningless, the coefficients for the Trade Freedom Index, the Investment Freedom Index, the Property Rights Index, and the Government Integrity Index are significantly positive.” What does this statement mean? Insignificant or significant?

Author: The problem has been fixed. Please refer to page 18, paragraph 2.

4. Can you justify why these findings are apparent? Provide some evidence-based rationale for these findings juxtaposing with previous studies.

“The neighboring countries' total economic freedom index demonstrates positive and significant effects. Furthermore, most economic freedom variables, such as the Business Freedom Index, the Investment Freedom Index, the Labor Freedom Index, the Trade Freedom Index, and the Government Integrity Index, exhibit a statistically significant and positive effect. However, neighboring countries' Monetary Freedom Index results in a decrease in the number of domestic arrivals. Among the various indicators, only improvements in the country's and neighboring countries' Investment Freedom Index and Government Integrity Index increase domestic arrivals. However, the Financial Freedom Index, the Tax Burden Index, and the Government Spending Index demonstrate no significant effect.”

Author: Explanations were added according to the opinion of the esteemed referee. Please refer to page 16 of the last paragraph, and page 17 of the first paragraph.

---

## [Decision Letter · Decision Letter 4]

25 Oct 2024

PONE-D-23-19081R4Economic Freedom Index Effects on Inbound Tourism in European Countries: A Spatial AnalysisPLOS ONE

Dear Dr. Hamza,

Thank you for submitting your manuscript to PLOS ONE. After careful consideration, we feel that it has merit but does not fully meet PLOS ONE’s publication criteria as it currently stands. Therefore, we invite you to submit a revised version of the manuscript that addresses the points raised during the review process.

**ACADEMIC EDITOR**The manuscript needs to be meticulously and carefully edited. It should provide a more comprehensive review of the literature and updated information. Some terms and theories should be explained more clearly such as "institutional economic theory"... The methods of analysis and techniques used need to be further clarified. The methodology should clearly explain a variables in model and the data processingl. It is particularly important for the article needs to clearly highlight the newly discovered points from the research. The format and language used are also important areas of concern for the article.

I am also enclosing the comprehensive comments from reviewer here for you to make necessary improvements to the article==============================

We look forward to receiving your revised manuscript.

Kind regards,

Thang Quyet Nguyen, Ph.D

Academic Editor

PLOS ONE

Journal Requirements:

Reviewers' comments:

Reviewer's Responses to Questions

**Comments to the Author**

1. If the authors have adequately addressed your comments raised in a previous round of review and you feel that this manuscript is now acceptable for publication, you may indicate that here to bypass the “Comments to the Author” section, enter your conflict of interest statement in the “Confidential to Editor” section, and submit your "Accept" recommendation.

Reviewer #9: All comments have been addressed

2. Is the manuscript technically sound, and do the data support the conclusions?

Reviewer #9: Yes

3. Has the statistical analysis been performed appropriately and rigorously? 

Reviewer #9: Yes

4. Have the authors made all data underlying the findings in their manuscript fully available?

Reviewer #9: Yes

5. Is the manuscript presented in an intelligible fashion and written in standard English?

Reviewer #9: Yes

6. Review Comments to the Author

Reviewer #9: 1. The abstract showed a study on the relationship between economic freedom and inbound tourism in European countries using spatial econometrics model. The study addressed an important research gap that need to conduct the research. The abstract mentioned methodology by using a spatial panel data approach and analyzing data from 41 European nations based on spatial framework. The abstract also showed clearly the key findings and policy implications.

2. The introduction provided a comprehensive between and well-structured entry into the topic of economic freedom index effecting on inbound tourism in European countries. The introduction mentioned an importance of inbound tourism in Europe, relevance of economic freedom, gaps in existing research, using spatial econometric models and clear research objectives. However, there are the following points should be improved:

- The introduction should not present the detail theories because the content will be showed in the literature review.

- Some terms and theories should be explained more clearly such as "institutional economic theory".

- Should briefly explain the spatial econometric model.

3. The literature review presented a comprehensive and well-structured content. The review showed the effects of economic freedom on inbound tourism. the determinants showed clearly such as GDP, price levels, exchange rates, income sensitivity. The review also mentioned spatial analysis and destination competitiveness theory. The literature review should clearly identify the gap based on the previous research and discussing more recent trend in research on inbound tourism. If possible the review should show a broader implications of economic freedom and clearly show the link between economic freedom and policy.

4. The methodology should clearly explain a variables in model and the data processing.

5. The results and finding are clearly and concisely.

6. Conclusions, policy implications and limitation:

- The conclusions presented the clearly the research topic.

- The policy implications should clearly suggest based on findings: the relationship between economic freedom and inbound tourism; tourism infrastructure, trade liberalization.

- Limitations should focus on model limitation (eg. not mention cultural and political factors), data constraints, external factors.

7. Revise the references and some paragraphs should be clarified and more focus.

7. PLOS authors have the option to publish the peer review history of their article (what does this mean? ). If published, this will include your full peer review and any attached files.

---

## [Author Response · Author response to Decision Letter 5]

31 Oct 2024

Author: Manuscript Number: PONE-D-23-19081R1

Title: Economic Freedom Index Effects on Inbound Tourism in European Countries: A Spatial Analysis

Authors' Response to the Editor and Reviewer's Comments

We are very grateful to the Editor and the respected reviewers for their valuable comments and suggestions which have improved the quality of our manuscript. In the revised manuscript, we have carefully addressed the comments and suggestions made by both the reviewers. The following is a summary of the main revisions that we have done in response to the Editor and reviewers' comments. The responses are highlighted in the revised version. The answers to comments are shown in blue below in this response letter, which also provides references to their placement in the manuscript.

Reviewer #9:

• The introduction should not present the detailed theories because the content will be shown in the literature review.

Author: We have revised the introduction section by removing the detailed theories and focusing on providing a general overview of the research topic. (Page 2, Paragraph 3)

• Some terms and theories should be explained more clearly such as "institutional economic theory".

Author: The term "institutional economic theory" has been clearly defined and elaborated upon to enhance understanding (Page 2, Paragraph 3)

• Should briefly explain the spatial econometric model.

Author: A concise explanation of the spatial econometric model has been added, outlining its key principles and relevance to the study (Page 4, Paragraph 1)

• The literature review should clearly identify the gap based on the previous research and discuss more recent trends in research on inbound tourism. If possible, the review should show a broader implication of economic freedom and clearly show the link between economic freedom and policy.

Author: The literature review has been expanded to explicitly identify gaps in existing research and to discuss recent trends in inbound tourism studies (Page 7, last Paragraph)

• The methodology should clearly explain variables in the model and the data processing.

Author: The methodology section now provides a detailed explanation of each variable included in the model (Page 8, Paragraph 2)

• The policy implications should clearly suggest based on findings: the relationship between economic freedom and inbound tourism; tourism infrastructure, trade liberalization.

Author: The policy implications section has been revised (Page 18, last Paragraph)

• Limitations should focus on model limitations (e.g., not mention cultural and political factors), data constraints, external factors.

Author: The limitations section has been refined to focus on model limitations, data constraints, and external factors (Page 19, Paragraph 1)

• Revise the references and some paragraphs should be clarified and more focused.

Author: All references have been updated and revised to ensure accuracy and relevance.

---

## [Decision Letter · Decision Letter 5]

6 Dec 2024

Economic Freedom Index Effects on Inbound Tourism in European Countries: A Spatial Analysis

PONE-D-23-19081R5

Dear Dr. Hamza,

We’re pleased to inform you that your manuscript has been judged scientifically suitable for publication and will be formally accepted for publication once it meets all outstanding technical requirements.

Kind regards,

Thang Quyet Nguyen, Ph.D

Academic Editor

PLOS ONE

---

## [Editor Report · Acceptance letter]

PONE-D-23-19081R5

PLOS ONE

Dear Dr. Hamza,

I'm pleased to inform you that your manuscript has been deemed suitable for publication in PLOS ONE. Congratulations! Your manuscript is now being handed over to our production team.

Kind regards,

on behalf of

Professor Thang Quyet Nguyen

Academic Editor

PLOS ONE